# PROCESS-VERIFIED REINFORCEMENT LEARNING FOR THEOREM PROVING VIA LEAN

**Minsu Kim**
KAIST AI
minsu_kim@kaist.ac.kr

**Se-Young Yun**
KAIST AI
yunseyoung@kaist.ac.kr

## ABSTRACT

While reinforcement learning from verifiable rewards (RLVR) typically has relied on a single binary verification signal, symbolic proof assistants in formal reasoning offer rich, fine-grained structured feedback. This gap between structured processes and unstructured rewards highlights the importance of feedback that is both dense and sound. In this work, we demonstrate that the Lean proof assistant itself can serve as a symbolic process oracle, supplying both outcome-level and fine-grained tactic-level verified feedback during training. Proof attempts are parsed into tactic sequences, and Lean's elaboration marks both locally sound steps and the earliest failing step, yielding dense, verifier-grounded credit signals rooted in type theory. We incorporate these structured rewards into a GRPO-style reinforcement learning objective with first-error propagation and first-token credit methods that balances outcome- and process-level advantages. Experiments with STP-Lean and DeepSeek-Prover-V1.5 show that tactic-level supervision outperforms outcome-only baselines in most settings, delivering improvements on benchmarks such as MiniF2F and ProofNet. Beyond empirical gains, our study highlights a broader perspective: symbolic proof assistants are not only verifiers at evaluation time, but can also act as process-level reward oracles during training. This opens a path toward reinforcement learning frameworks that combine the scalability of language models with the reliability of symbolic verification for formal reasoning.

## 1 INTRODUCTION

Automated theorem proving (ATP) is one of the long-term goals of AI (Newell et al., 1957). Compared to reasoning in natural language (NL), which often contains vague or ambiguous symbols, formal theorem proving based on formal logic and type theory provides technical and precise language for proving mathematical theorem (Church, 1940; Fitting, 1996). Currently, interactive theorem provers (ITP) such as Lean (de Moura et al., 2015; Moura & Ullrich, 2021), Isabelle (Nipkow et al., 2002) and Coq (Barras et al., 1997), serve as reliable and powerful tools for verifying mathematical proofs. Lean proofs are sequences of tactics, with automation handling routine arithmetic/logic and verification-so ITPs provide a middle ground between full automation and human guidance

By contrast, LLMs model next-token probabilities from large corpora via pre- and post-training, learning lexical correlations rather than rule-based symbolic manipulation (Brown et al., 2020). With further techniques such as instruction tuning and Reinforcement Learning from Human Feedback (RLHF), LLMs have evolved to handle a wide range of tasks, including question answering, summarization, dialogue (Ouyang et al., 2022; Bai et al., 2022). In particular, reinforcement learning (RL) approaches for reasoning tasks aim to enhance the model's reasoning ability by encouraging the generation of long chains of thought rationale (DeepSeek-AI et al., 2025; OpenAI et al., 2024).

Compared to other reasoning tasks which often verify or reward LLMs' response according to its final answer (Cobbe et al., 2021), the theorem prover can verify the correctness of entire proof when LLMs respond with formal language. In this context, given the human-in-the-loop nature of ITPs, there have been growing attempts to use LLMs for formal theorem proving tasks (AlphaProof and AlphaGeometry teams, 2024; Trinh et al., 2024). LLMs act as prover agents while theorem provers serve as verifiers, being used either at inference time-to search and validate tactics and premises-or for augmenting formal reasoning datasets with verified samples (Lample et al., 2022; Wang et al.,

2023; Ying et al., 2024a; Zhu et al., 2025). Furthermore, some recent studies incorporated binary feedback from the Lean theorem prover into its online RL framework (Xin et al., 2024b).

The tactic-based proof structure in Lean contains information relevant for reasoning tasks such as the positions of tactics or the nature of proof errors or failures. This structured information captures not just the outcome of a proof, but also the underlying reasoning process. However, despite its potential, only a few works have explored incorporating this kind of fine-grained supervision into the training of LLMs (Ji et al., 2025). At the same time, recent RL approaches for reasoning have increasingly emphasized the use of process-based reward models (PRMs) to guide model behavior. While these models show promising performance, there is still a lack of clarity around how PRMs are constructed, how the reasoning step or step reward should be defined, what training signals or datasets they should depend on (Yuan et al., 2024; Luo et al., 2024; Cui et al., 2025).

Unlike recent approaches that rely on PRMs or long NL CoT (Lin et al., 2025a;b), we directly leverage the Lean proof assistant as a *symbolic process oracle* during RL training, without any natural-language guidance. For each generated proof, Lean provides (i) a global outcome signal and (ii) fine-grained tactic-level feedback via info trees and error logs.

While fine-grained tactic level signals are available from Lean, leveraging them effectively during RL training is nontrivial. Lean outputs symbolic, tree-structured language feedback, such as proof states and error locations, whereas LLMs operate over autoregressive token sequences and learn from scalar rewards in RL. This representational mismatch creates a credit-assignment challenge: symbolic verifier feedback must be transformed into structured token-level training signals. To bridge this gap, we introduce a structured credit assignment framework for integrating symbolic verifier signals into an online RL objective, requiring three components: (i) a formulation for incorporating fine-grained signals, (ii) a principled rule for reducing Leans symbolic feedback into per-tactic scores, and (iii) a mapping from per-tactic scores to token-level advantages. We instantiate this pipeline using a tactic-level MDP, a first-error propagation rule grounded in Leans semantics, and a first-token credit assignment strategy.

We integrate the resulting per-tactic signals into a Group Relative Policy Optimization (GRPO) style objective combining outcome- and process-level advantages. This enables precise, type-theoretic credit assignment grounded in verifier feedback without the need for an auxiliary PRM. Empirically, we found that incorporating symbolic verifier feedback into the RL objective consistently improves performance on MiniF2F and ProofNet, demonstrating the value of fine-grained verifier signals for reliable credit assignment in reasoning tasks. Our key contributions are as follows:

- **Formalizing Lean's Symbolic Feedback.** We formalize Leans symbolic, tactic-level feedback and reduce it into scalar training signals that enable fine-grained, token-level credit assignment.

- **Symbolic verifier-guided RL.** We integrate outcome and tactic-level rewards derived from Lean into an RL framework, providing dense and verifiable credit assignment.

- **Stable improvements on benchmarks.** On MiniF2F and ProofNet, our approach consistently outperforms both outcome-only RL and vanilla baselines, yielding more stable and robust gains without NL notation or external PRM.

## 2 RELATED WORK

**Automatic Theorem Proving** An automated theorem prover typically consists of two stages. The first is autoformalization, i.e., translating natural language mathematical statements into formal ones. LLMs have been used for this task (Wu et al., 2022), producing datasets such as MiniF2F, ProofNet, Deepseek-Prover, and LeanWorkbook (Zheng et al., 2022; Azerbayev et al., 2023; Xin et al., 2024a; Ying et al., 2024a). The second stage is proof generation, which can be performed step-by-step via tree search (Polu & Sutskever, 2020; Azerbayev et al., 2024; Wu et al., 2024; Xin et al., 2024b) or by generating entire proofs at once (Xin et al., 2024a; Lin et al., 2025b). Existing approaches such as Lean-STaR and RMaxTS use Lean only as a step-checker during inference (Lin et al., 2025a; Xin et al., 2024b), whereas recent work has employed Lean as a whole-proof verifier during training (Wang et al., 2025a; Zhang et al., 2025; Ren et al., 2025). In this paper, we go beyond step-checking or whole-proof verification by using Leans fine-grained, tactic-level feedback as process-based rewards in online RL training.

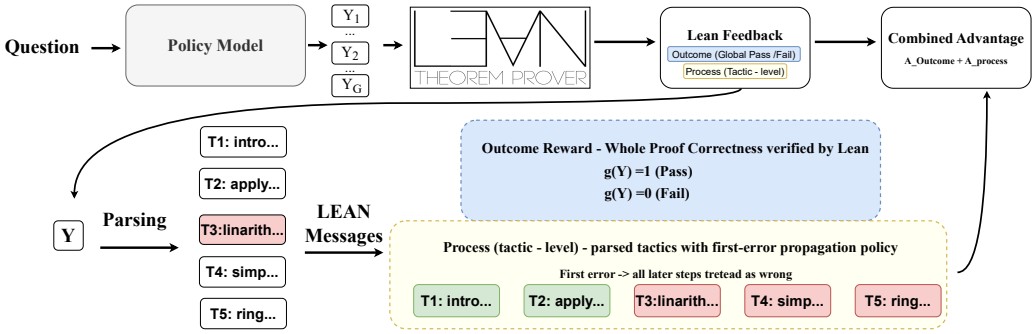

Figure 1: Overall framework for combining outcome and tactic level rewards via Lean: the proof $Y$ is parsed into tactics $T_1, \ldots, T_5$, with Lean providing outcome feedback $g(Y)$ and step-level errors (e.g., failure at $T_3$ invalidates later tactics). Rewards are then assigned to the first token of each tactic.

**Reinforcement Learning in Language Models** Beyond algorithmic advances such as PPO (Schulman et al., 2017) and GRPO (Shao et al., 2024), reward shaping and credit assignment remain core challenges in RL. Outcome-based rewards (Cobbe et al., 2021), though widely used in RLHF, suffer from sparsity (Chan et al., 2024; Zheng et al., 2023). Process-based reward models (PRMs) address this by assigning step-level rewards (Lightman et al., 2023; Setlur et al., 2024; Kazemnejad et al., 2024; Yuan et al., 2024; Cui et al., 2025). Rewards can be defined implicitly (Cui et al., 2025) or explicitly via correctness annotations (Lightman et al., 2023) or Monte Carlo rollout success rates (Wang et al., 2024), but existing methods require large annotated datasets of step-level correctness. This motivates our approach of leveraging the Lean prover itself as a process oracle, automatically verifying each step without human labels or sampling. Additional discussion is in Appendix A.

## 3 PRELIMINARIES

### 3.1 LEAN4

In Lean theorem proving, a statement to be established is represented as an initial goal and incrementally reduced into subgoals through a sequence of tactics. Each tactic is parsed and elaborated by unifying it with lemmas or theorems in the library, generating new subgoals, and verifying their validity. The elaboration stage produces structured info trees that record proof states and error messages. Finally, the kernel ensures that the elaborated proof is type-theoretically consistent and constitutes a valid proof for the original theorem.

Formally, let $x$ denote a theorem statement provided to an LLM, and let $Y$ be the response, a proof expressed in the Lean language. Write $\mathcal{Y}$ for the set of Lean proofs and $\mathcal{T}$ for the set of tactics. For $Y \in \mathcal{Y}$, We view a proof $Y$ as a sequence of tactics $(T_1, T_2, \ldots, T_{N(Y)})$ parsed from the Abstract Syntax Tree (AST) and sorted by their starting positions, where $N(Y)$ is the number of tactics in $Y$ which aligns with the LLM's autoregressive generation process. Each tactic $T_i$ comprises corresponding tokens $y_t$ in $Y$. Lean represents tactics as AST nodes; each node encodes the tactics syntactic structure and binding context, and may carry metadata such as error messages, proof states, and an index through which users (or training frameworks) can interact with Lean. If a tactic does not appear in the error log, then it has been elaborated successfully and passed Lean's internal rule-based verification, which guarantees that the step is locally sound under dependent type theory. Thus, any tactic not marked as an error constitutes a verified reasoning step-even if it does not contribute to closing the proof because some subgoals remain or later tactics fail. In other words, Lean ensures tactic-level soundness, while proof-level completeness depends on whether the entire sequence resolves all goals. Leveraging this parsing and validation feedback, we define the parsing function $f : \mathcal{Y} \to \mathcal{T}^*$ to be the sequence obtained by sorting $\mathsf{TacSet}(Y)$: $f(Y) = (T_1, \ldots, T_{N(Y)})$. We also define the global scoring function $g : \mathcal{Y} \to [0, 1]$, where $g(Y) = 1$ if $Y$ passes the Lean verifier and 0 otherwise, and the per-tactic scoring function $\varphi : \{(Y, T) \mid Y \in \mathcal{Y}, T \in \mathsf{TacSet}(Y)\} \longrightarrow \{1, d_1, d_2\}$.

Specifically,

$$\varphi(Y, T) = \begin{cases} 1, & \text{if } g(Y) = 1, \\ d_1, & \text{else if } g(Y) = 0 \text{ and } T \text{ has no errors in Lean,} \\ d_2, & \text{else if } g(Y) = 0 \text{ and } T \text{ contains errors.} \end{cases}$$

Combining these components, we represent Lean's role via $f, g, \varphi$ as

$$\text{Lean}: \ \mathcal{Y} \to \{0, 1\} \times (\mathcal{T} \times \{1, d_1, d_2\})^*,$$
$$\text{Lean}(Y) = \big(g(Y), [(T_1, \varphi(Y, T_1)), \ldots, (T_{N(Y)}, \varphi(Y, T_{N(Y)}))]\big)_{f(Y) = (T_1, \ldots, T_{N(Y)})}.$$

## 3.2 TACTIC-LEVEL MDP

We define a tactic-level Markov Decision Process (MDP) as the tuple $\mathcal{M} = (\mathcal{S}, \mathcal{A}, r, F, m)$. The state space $\mathcal{S}$ contains partial formal proofs; each $s \in \mathcal{S}$ is the proof prefix produced so far. The action space $\mathcal{A}$ coincides with the tactic space $\mathcal{T}$; each action $a \in \mathcal{A}$ is a single Lean tactic. The reward function $r : \mathcal{S} \times \mathcal{A} \to \mathbb{R}$ assigns a tactic-level reward $r(s, a)$. The transition function $F : \mathcal{S} \times \mathcal{A} \to \mathcal{S}$ is deterministic: $s_{j+1} = F(s_j, a_j) = s_j \oplus a_j$, where $\oplus$ denotes concatenation of the tactic $a_j$ to the proof $s_j$ at time step $j$. Transitions are pure concatenations; Lean feedback affects $r$, not $F$. Let $\mathcal{S}_{\text{term}} \subseteq \mathcal{S}$ be EOS absorbing states. Let $m \in \mathcal{S}$ be the initial state. In Section 4, we extend this formulation with outcome- and tactic-level rewards derived from the Lean theorem prover to obtain the final training signal.

## 3.3 CREDIT ASSIGNMENT IN REINFORCEMENT LEARNING

PPO assigns a sparse end-of-sequence reward and propagates credit with a value model with Generalized Advantage Estimate (GAE), reducing variance at the cost of extra learning complexity; full details are deferred to Appendix E.

In contrast, REINFORCE style GRPO optimizes directly from verifiable whole-trajectory rewards without a value model. For a prompt $q$, we sample $G$ responses $\{y_i\}_{i=1}^G$ from $\pi_{\text{old}}$ and obtain rewards $r_i$. A normalized, response-level advantage is applied uniformly to all tokens of $y_i$:

$$\hat{A}_i = \frac{r_i - \text{mean}(r)}{\text{std}(r)}.$$

The objective is

$$L_{\text{GRPO}}(\theta) = \mathbb{E}\Big[\frac{1}{G} \sum_{i=1}^{G} \big\{ \min\big(\frac{\pi_\theta(y_i \mid q)}{\pi_{\theta_{\text{old}}}(y_i \mid q)} \, \hat{A}_i, \ \text{clip}\big(\frac{\pi_\theta(y_i \mid q)}{\pi_{\theta_{\text{old}}}(y_i \mid q)}, 1-\epsilon, 1+\epsilon\big) \hat{A}_i\big) - \beta \, D_{\text{KL}}[\pi_\theta \| \pi_{\text{ref}}]\big\}\Big].$$

We make this dense and sound by injecting *Lean-derived tactic advantages* into GRPO: the outcome signal remains at response level, while tactic-level signals are mapped to tokens at the first token of each tactic (Sec. 4). This preserves GRPO's simplicity while addressing sparse credit.

## 4 METHOD

### 4.1 DEFINE TACTIC-LEVEL REWARDS

In the previous section, we modeled the correctness of proofs $Y$ generated by the Lean proof assistant and parsed and verified each tactic within $Y$. We now introduce a reward mechanism that integrates both outcome-based and process-based signals explicitly into the RL framework. Specifically, we employ an outcome-based reward defined through a function $g(Y)$, similar to approaches used by (DeepSeek-AI et al., 2025), as a global reward evaluating the entire proof. Additionally, we define a process-based reward $\varphi(Y, T)$, assessing the correctness or validity at the level of individual tactics $T \in Y$. Unlike implicit rewards or Monte Carlo estimations typically interpreted as process rewards, our method explicitly assigns correctness-based rewards at each tactic step.

Assume that, analogous to the GRPO training rollout framework, given a question $q$, an LLM generates a group of responses $\{Y_1, Y_2, \ldots, Y_G\}$. Lean produces an outcome-based rewards:

$$r_{\text{outcome}}(Y_i) = g(Y_i)$$

We define the outcome-based advantage for any token $y_{i,t}$ in response $Y_i$ as:

$$A_{\text{outcome}, i, t} = \frac{g(Y_i) - \text{mean}\big(g(Y_1), \ldots, g(Y_G)\big)}{\text{std}\big(g(Y_1), \ldots, g(Y_G)\big)}.$$

Beyond binary outcome verification signals, we further design elaborate rewards based on the AST feedback produced by the Lean parser as in section 3.1. We leverage this AST feedback to distinguish between different kinds of tactics: for example, whether a tactic is elaborated successfully (i.e., type-correct and locally sound), but may still leave unresolved subgoals that prevent the proof from being completed, or whether it has type errors or parser-level mismatches. This structured feedback allows us to assign more fine-grained process-based rewards. Since, we sorted the tree node containing proof state by increasing order, we apply a First Error Propagation rule when mapping Lean's feedback into tactic-level rewards as (Lu et al., 2024; Lightman et al., 2023). Given a sequence of tactics $(T_1, \ldots, T_N)$, once an error is observed at $T_j$, we propagate this failure to all subsequent tactics, i.e., every $T_k$ with $k \geq j$ is treated as erroneous for the purpose of reward assignment.

$$\text{Let } j = \min\{i : T_i \text{ contains an error}\}. \quad \varphi(Y, T_k) = \begin{cases} 1, & g(Y) = 1. \\ d_1, & g(Y) = 0 \text{ and } k < j \text{ and no error}, \\ d_2, & g(Y) = 0 \text{ and } k \geq j, \end{cases}$$

Unlike Lean, which parses proofs into a tree structure, the LLM generates proofs in an autoregressive, causal manner. Once the first erroneous tactic $T_j$ occurs, the continuation $T_{j+1}, \ldots, T_N$ is conditioned on an invalid prefix, and therefore cannot constitute a valid reasoning process. First-error propagation enforces this principle by assigning error signals to all subsequent tactics, ensuring causal and type-theoretic credit assignment.

For any arbitrary response $Y_i$, composed of tactics $Y_i = \{T_{i,1}, T_{i,2}, \ldots\}$, if we set $s_j, a_j$ as the state and tactic $T_{i,j}$ at step $j$ in response $Y_i$, the process-based reward for tactic $T_{i,j}$ is:

$$r_{\text{process}}(s_j, a_j) = r_{\text{process}, i, j} = \varphi(Y_i, T_{i,j}).$$

The corresponding process-based advantage is

$$A_{\text{process}, i, j} = r_{\text{process}, i, j} - \text{mean}(g(Y_1), \ldots, g(Y_G)).$$

Here, the subtraction of the mean outcome reward serves as a dynamic baseline reflecting the difficulty of the problem $q$ as GRPO algorithm. If the problem is easier, the mean outcome reward becomes higher, thus penalizing incorrect proofs and their tactics more heavily. Conversely, for more challenging problems, the lower baseline imposes less severe penalties.

## 4.2 INTEGRATING LEAN INTO TACTIC-BASED REINFORCEMENT LEARNING

We then integrate these two types of advantages into the standard GRPO objective as follows.

$$A_{i,t} = A_{\text{outcome}, i, t} + \mathbf{1}\{t = \text{first}(T_{i,s(i,t)})\} \cdot A_{\text{process}, i, s(i,t)},$$

where $s(i,t) \in \{1, \ldots, N\}$ is the index of the tactic containing the token $t$ in $Y_i$, $\text{first}(T_{i,j})$ indicates the first token of the tactic. i.e., we assign the tactic advantage only to the first token of each tactic. We applied the advantage $A_{i,t}$ into GRPO objective function:

$$
\begin{aligned}
L(\theta) = \mathbb{E}_{q \sim P(Q), \, \{Y_i\}_{i=1}^G \sim \pi_{\theta_{\text{old}}}(Y|q)} \\
\left[ \frac{1}{G} \sum_{i=1}^G \left\{ \frac{1}{|Y_i|} \sum_{t=1}^{|Y_i|} \min\left( \rho_{i,t} A_{i,t}, \, \text{clip}\left(\rho_{i,t}, 1-\epsilon, 1+\epsilon\right) A_{i,t} \right) - \beta D_{\text{KL}}\left[\pi_\theta \| \pi_{\text{ref}}\right] \right\} \right].
\end{aligned}
\tag{1}
$$

where $\rho_{i,t} = \frac{\pi_\theta\left(y_{i,t}|q, Y_{i,<t}\right)}{\pi_{\theta_{\text{old}}}\left(y_{i,t}|q, Y_{i,<t}\right)}$. This formulation explicitly leverages both the global correctness signal $A_{\text{outcome}, i, t}$ from proof outcomes and the detailed, tactic level correctness assessment $A_{\text{process}, i, s(i,t)}$. By combining them into a single advantage $A_{i,t}$, we enrich the learning signal provided to the LLM based proof generator under the GRPO framework.

In Appendix J, we provide a mathematical grounding and interpretation of our method via a potential-based reward shaping. We can interpret our credit assignment as a discrete approximation of potential-based reward shaping, where the potential function of a state is defined as the probability that the current proof prefix can be completed into a valid Lean proof. We treat the error state as an absorbing state and set its potential to zero (see Appendix J for details).

Rather than propagating cumulative rewards across an entire proof trajectory, we collapse credit assignment to Lean-verified, tactic-level signals. In general RL, a suboptimal step may still obtain positive return if later rewards are high, but in mathematical proof, this could be unsound: once a

| Model | Model size | Budget [1] | MiniF2F-Test | ProofNet-Test |
|---|---|---|---|---|
| **Whole-Proof Generation Methods** | | | | |
| DeepSeek-Prover-V1.5-SFT (Xin et al., 2024a) | 7B | 32 | $46.2\% \pm 0.2$ | $14.3\% \pm 0.3$ |
| | | 64 | $47.5\% \pm 0.1$ | $15.05\% \pm 1$ |
| DeepSeek-Prover-V1.5-RL (Xin et al., 2024a) | 7B | 32 | $48\% \pm 0$ | $16\% \pm 1$ |
| | | 64 | $48.8\% \pm 0.4$ | $17.4\% \pm 0.6$ |
| Goedel-Prover-SFT (Lin et al., 2025c) | 7B | 32 | $56.9\% \pm 0.4$ | $15.6\% \pm 0.5$ |
| | | 64 | $57.9\% \pm 0.5$ | $16.7\% \pm 0$ |
| STP-Lean (Dong & Ma, 2025) | 7B | 32 | $55.9\% \pm 0.2$ | $17.2\% \pm 0$ |
| | | 64 | $56.7\% \pm 0.2$ | $\mathbf{19.1\%} \pm 0.4$ |
| **STP-Lean + Ours** | 7B | 32 | $\mathbf{57.1\%} \pm 0.8$ | $18.6\% \pm 0.3$ |
| | | 64 | $\mathbf{59.2\%} \pm 0.5$ | $19\% \pm 0.3$ |
| DeepSeek-Prover-V1.5 + STP | 7B | 32 | $54.9\% \pm 0.7$ | $16.8\% \pm 0.3$ |
| | | 64 | $57.2\% \pm 0.2$ | $17.7\% \pm 0$ |
| **DeepSeek-Prover-V1.5 + STP + Ours** | 7B | 32 | $\mathbf{56.3\%} \pm 0.6$ | $\mathbf{17.6\%} \pm 0.8$ |
| | | 64 | $\mathbf{57.8\%} \pm 0.4$ | $\mathbf{18.5\%} \pm 0.3$ |
| **Tree Search Methods** | | | | |
| Lean-STaR | 7B | $64 \times 1 \times 50$ | $46.3\%$ | – |
| InternLM2-Math-Plus-7B (Ying et al., 2024b) | 7B | $1 \times 32 \times 100$ | $48.8\%$ | – |
| InternLM2.5-StepProver | 7B | $4 \times 32 \times 600$ | $58.5\% \pm 0.9$ | – |
| DeepSeek-Prover-V1.5-RL + RMaxTS (Xin et al., 2024a) | 7B | 3,200 | $55.0\% \pm 0.7$ | $21.5\% \pm 0.8$ |

Table 1: Budgets for whole-proof methods denote the *sample budget* ($N$) per problem; for tree-search methods, budgets denote the authors reported *search expansions counts*. We compare with InternLM family and DeepSeek-Prover based tree search methods for fair comparison with our method. Bold indicates the best number within the whole-proof block. All our GRPO-style runs use the same STP subset, generations per query, and a 15s Lean timeout. The notation $\mu \pm \sigma$ indicates the mean and the standard deviation each.

tactic fails, all subsequent steps are invalid under first-error propagation. Empirically, return-based credit led to unstable optimization, as it requires a value function or auxiliary estimator to normalize scale and reduce variance. Hence, we adopt a simpler formulation that combines normalized outcome-level signals with tactic-level rewards, without computing returns (See Appendix G).

## 5 EXPERIMENTS

### 5.1 EXPERIMENTAL SETUP

We trained on 10k samples randomly drawn from the STP dataset (3.26M proofs). Proofs were verified via Lean through a REPL interface, with a 15s timeout per attempt. Baselines included STP-Lean and DeepSeek-Prover-V1.5-SFT, the latter additionally fine-tuned on 500k STP samples before RL. We used non-CoT prompt, response styles as in (Xin et al., 2024b). We used tactic-level rewards $d_1 = -0.05$ and $d_2 = -0.1$ for the main experiment. Full hyperparameters and training details are provided in Appendix B.

### 5.2 MAIN RESULTS

In Table 1, the results on both the MiniF2F and ProofNet datasets demonstrate that training with tactic-based advantage via Lean consistently enhances model performance across most evaluation settings. For the STP-Lean model, our method improves MiniF2F performance up to $+2.5\%$p (pass@64), and ProofNet performance by $+1.4\%$p (pass@32), while showing a negligible decrease of $-0.1\%$p on pass@64. Similarly, for DeepSeek-Prover-V1.5, our approach achieves marginal yet consistent increases across all benchmarks.

---

[1]**Budgets are not directly comparable:** tree-search budgets count expansions/verifier calls at inference, whereas our budgets count whole-proof samples. Our aim is to improve single-shot generation under a different compute regime.

| Model | Model Size | Budget | MiniF2F - Test | ProofNet - Test |
|---|---|---|---|---|
| STP + Outcome only (GRPO) | 7B | 32 | $55.7\% \pm 1$ | $17.4\% \pm 0.6$ |
| | | 64 | $57.9\% \pm 0.5$ | $\mathbf{19\%} \pm 0.3$ |
| STP + Tactic only | 7B | 32 | $55.6\% \pm 0.6$ | $18.3\% \pm 0$ |
| | | 64 | $56.8\% \pm 0.6$ | $17.9\% \pm 0.8$ |
| STP + Outcome+Tactic RL (ours) | 7B | 32 | $\mathbf{57.1\%} \pm 0.8$ | $\mathbf{18.6\%} \pm 0.3$ |
| | | 64 | $\mathbf{59.2\%} \pm 0.5$ | $\mathbf{19\%} \pm 0.3$ |
| DeepSeek-Prover-V1.5 + Outcome only (GRPO) | 7B | 32 | $55.3\% \pm 0.4$ | $16.8\% \pm 0.8$ |
| | | 64 | $57.4\% \pm 0.4$ | $17.6\% \pm 0.8$ |
| DeepSeek-Tactic only | 7B | 32 | $54.9\% \pm 0.7$ | $16.8\% \pm 0.8$ |
| | | 64 | $57.8\% \pm 1$ | $17.6\% \pm 0.3$ |
| DeepSeek-Prover-V1.5 + Outcome+Tactic RL (ours) | 7B | 32 | $\mathbf{56.3\%} \pm 0.6$ | $\mathbf{17.6\%} \pm 0.8$ |
| | | 64 | $\mathbf{57.8\%} \pm 0.4$ | $\mathbf{18.5\%} \pm 0.3$ |

Table 2: Ablation study of STP-Lean with various verifier methods on MiniF2F-Test and ProofNet-Test benchmarks.

Across both MiniF2F and ProofNet, leveraging Lean as a *process-level oracle* yields consistent, stable gains over outcome-only reinforcement learning, without increasing training cost. In particular, in Table 2, when applied to DeepSeek-Prover models, GRPO fails to yield any gains on the ProofNet-Test set, and in some cases even underperforms relative to the supervised baseline. This highlights a key limitation of purely outcome-based credit assignment: it often lacks stability and fails to provide consistent guidance for proof search.

By comparison, tactic-level credit assignment yields more reliable improvements. While minor drops appear in some settings, it generally provides stable gains over outcome-only GRPO. For example, on MiniF2F (pass@64), STP-Lean + Ours improves by +2.5%p over the baseline, compared to +1.2%p with GRPO. As shown in Table 1 and 2, tactic-based training consistently matches or surpasses both the supervised baseline and GRPO. Importantly, this stability comes with almost no extra cost: since both methods already use REPL interactions with Lean, the additional sorting and scoring overhead is negligible.

Compared to strong search-based baselines (e.g., InternLM families, DeepSeek-Prover-RL+RMaxTS), our single-shot, whole-proof training approaches their reported accuracy (e.g., 59.2% vs. 58.5% pass@64 on MiniF2F) while avoiding large search-time compute.

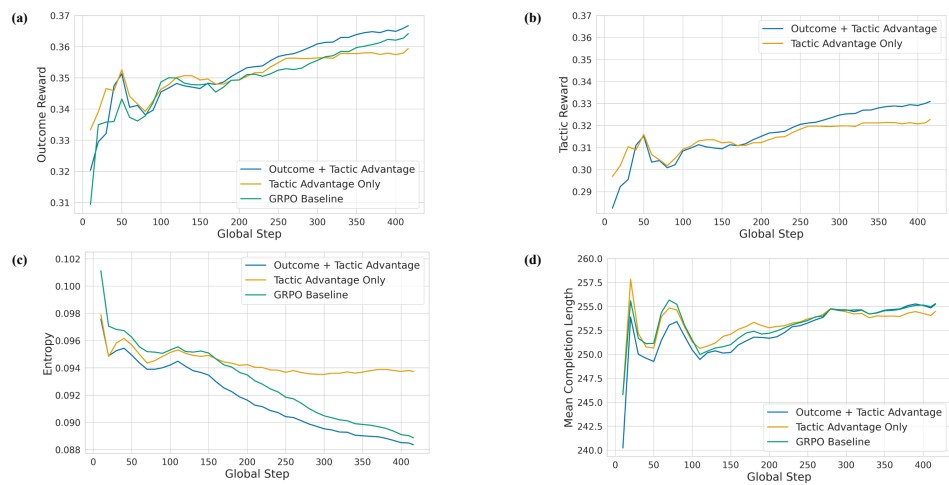

Figure 2: Training dynamics showing (a) outcome reward, (b) tactic reward, (c) entropy, and (d) mean of response length during reinforcement learning. We used time-weighted smooting with smoothing factor 0.99.

## 5.3 ANALYSIS

**The Role of Outcome and Tactic Rewards.** Integrating both outcome-level and tactic-level signals yields more effective learning than employing either signal in isolation. Outcome-only RL, as in GRPO, is constrained by the sparsity of binary feedback: improvements are gradual and the final performance plateaus at a relatively low level (Figure 2(a)). In contrast, tactic-only training provides dense feedback but lacks a global objective, resulting in premature convergence. When combined, outcome rewards serve as a global objective function, while tactic rewards provide local credit assignment, enabling both rapid progress and higher performance. This complementary relationship is further reflected in Figure 2(b), where tactic-only supervision's tactic reward plateaus, but outcome-tactic combined rewards continue to increase steadily. The results in Table 2 supports this finding: outcome signals enforce proof-level correctness, while tactic signals supply verifiable intermediate feedback; only their integration consistently improves performance across benchmarks.

**Entropy and Proof Length.** The use of fine-grained rewards influences exploration not by indiscriminately broadening the search space but by focusing learning on more informative decision points. As shown in Figure 2(c), outcome+tactic training converges to lower entropy than tactic-only and outcome-only settings, indicating that the policy becomes more decisive as training progresses. This does not correspond to mode collapse: Figure 2(d) shows that the average proof length remains stable across all methods, suggesting that the performance gains are not attributable to trivial lengthening of outputs. Instead, denser intermediate rewards appear to reduce the need for broad stochastic exploration, guiding the model toward more efficient proof strategies.

**Tactic to Token Level Credit Assignment.** After defining tactic-level rewards, next step is how to distribute them across tokens. In our main method, the tactic advantage is assigned only to the first token of the tactic. For comparison, we conducted ablations where the tactic advantage was instead (i) distributed to all tokens of a tactic, (ii) assigned only to the last token, (iii) keep first token reward distribution, but additionally choose 10% tokens within the tactic with respect to high entropy. As Wang et al. (2025d) showed that high entropy tokens could be reasoning drive tokens, we speculated that this method can automatically select the tokens for serving as fork in formal reasoning. Assigning credit to the first token of each tactic achieves the most stable and consistent improvements, as evidenced by Table 3. Alternative strategies do not yield comparable gains and in some cases even degrade performance. This outcome aligns with the semantics of Lean proofs: the first token corresponds to the tactic keyword (e.g., `intro`, `apply`, `have`), determining the subsequent proof strategy and constrains the structure of subgoals. Concentrating credit on this decision point enhances the models ability to select tactics appropriately, resulting in more reliable downstream reasoning. This finding is also aligned with (Fang et al., 2025), showing that focusing on key tokens during training improves performance on long-context tasks.

**Reward Strategy for Tactic-level Feedback.** For tactic-level feedback to be effective, it must reflect the sequential dependency of proof construction, account for task difficulty, and distinguish between partially correct and erroneous steps. The first-error propagation rule ensures that once an error occurs, subsequent tactics are treated as invalid; removing this rule significantly reduces performance (Table 4), because once the first error occurs, the remaining tactics are evaluated in an invalid context and cannot salvage correctness. Incorporating a difficulty-normalized baseline further stabilizes training, while its absence leads to degraded results. Finally, differentiating penalties between partially correct tactics and outright erroneous ones proves essential: collapsing these into a single penalty $d_1 = d_2$ yields inconsistent outcomes- improvements on MiniF2F but declines on ProofNet. These results indicate that an effective tactic-level reward scheme must combine sequential error propagation, difficulty-aware normalization, and differentiated penalties in order to provide stable and semantically faithful learning signals. In the sensitivity analysis of Appendix C, assigning different values to $d_1$ and $d_2$ leads to robust performance, tending to outperform the GRPO baseline and yielding the strongest improvements on MiniF2F.

**Effect of Verification Timeouts** When using Lean as a verifier, long proofs can lead to excessive verification time, so we introduced timeout thresholds of 5, 10, 15, and 30s (Figure 3). A 5s limit gave the worst results, since even relatively simple proofs often exceeded this window and produced too few valid reward signals. In contrast, 10-30s yielded much stronger performance, with 15s giving the best overall balance. Interestingly, 10-15s sometimes outperformed 30s despite the shorter allowance. We attribute this to the fact that discarding overly complex proofs biases training

| Model | Model Size | Sample Budget | MiniF2F - Test | ProofNet - Test |
|---|---|---|---|---|
| All tokens | 7B | 32 | $56.3\% \pm 0.6$ | $18.1\% \pm 0.8$ |
|  |  | 64 | $57.8\% \pm 0.7$ | $18.1\% \pm 0.8$ |
| Entropy-based | 7B | 32 | $56.4\% \pm 0.2$ | $17.9\% \pm 0.8$ |
|  |  | 64 | $57.1\% \pm 0.5$ | $18.5\% \pm 0.3$ |
| Last token | 7B | 32 | $56.7\% \pm 0.9$ | $17.2\% \pm 0$ |
|  |  | 64 | $57.5\% \pm 0.6$ | $17.7\% \pm 0.5$ |
| First token | 7B | 32 | $\mathbf{57.1\%} \pm 0.8$ | $\mathbf{18.6\%} \pm 0.3$ |
|  |  | 64 | $\mathbf{59.2\%} \pm 0.5$ | $\mathbf{19\%} \pm 0.3$ |

Table 3: Ablation study of STP-Lean on how to distribute tactic-level advantages across tokens.

| Model | Model Size | Sample Budget | MiniF2F - Test | ProofNet - Test |
|---|---|---|---|---|
| No First Error | 7B | 32 | $56.4\% \pm 0.9$ | $17.4\% \pm 0.3$ |
|  |  | 64 | $58.2\% \pm 0.7$ | $18.3\% \pm 0.3$ |
| No Baseline | 7B | 32 | $56.7\% \pm 0.2$ | $17.9\% \pm 0.3$ |
|  |  | 64 | $57.4\% \pm 0.7$ | $18.3\% \pm 0.5$ |
| Same tactic reward | 7B | 32 | $\mathbf{57.7\%} \pm 0.2$ | $17.6\% \pm 0.6$ |
|  |  | 64 | $58.7\% \pm 0.8$ | $18.1\% \pm 0.6$ |
| Outcome+Tactic RL (ours) | 7B | 32 | $57.1\% \pm 0.8$ | $\mathbf{18.6\%} \pm 0.3$ |
|  |  | 64 | $\mathbf{59.2\%} \pm 0.5$ | $\mathbf{19\%} \pm 0.3$ |

Table 4: Ablation study on reward strategies for tactic-level feedback in STP-Lean. Additional experiments include removing the first-error propagation policy (No First Error), removing the baseline extraction (No Baseline). and using equal penalties for all tactics (Same tactic reward).

toward shorter and more efficient proof strategies. This effect is amplified in our setting because we evaluate non-CoT responses purely by Lean verification (without NL commentary): longer outputs are not only slower to check but also more error-prone. As a result, shorter verification limits encourage the model to generate concise, canonical proofs, which we hypothesize leads to better generalization at test time.

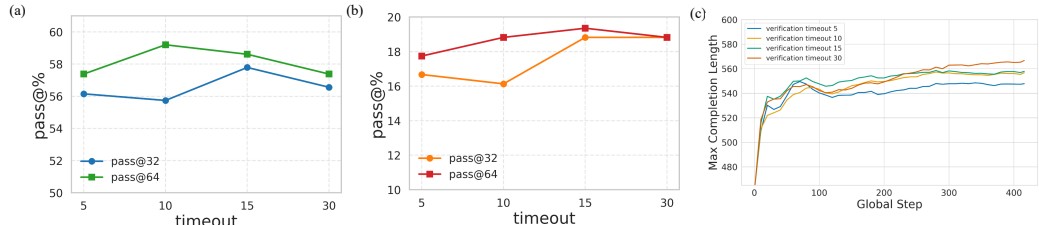

Figure 3: Ablation study of STP-Lean on different Lean verification timeouts (5, 10, 15, and 30 seconds) during outcome+tactic based training. We report evaluation performance on the MiniF2F and ProofNet benchmarks (a),(b), and the maximum response length observed during training (c).

**Qualitative Analysis** We conduct a qualitative analysis to better understand the differences between our tactic-reward-based approach and the baseline STP model. Specifically, we examine proofs from two benchmark problems: Imo_1960_p2 in the MiniF2F benchmark and 1_14 from ProofNet (See Appendix F). Table 8 presents the proof generated by our tactic-reward model, while Table 9 shows the corresponding proof from the STP model.The key difference in the first example lies in how the upper bound $x < 45/8$ is established. The STP model attempts to use the nonlinear inequality tactic nlinarith, which results in an error. By contrast, our tactic-reward model learns to penalize such invalid tactic choices. Instead, it carefully applies previously proven assumptions and intermediate lemmas before invoking nlinarith, thereby producing a correct and more robust proof. The second example comes from the ProofNet benchmark (1_14 exercise). As shown in Table 10, the tactic-reward model begins by normalizing the problem using a rewrite tactic. In contrast, the baseline model in Table 10 skips this normalization step and directly attempts inequality manipula-

tions, which ultimately causes the proof to fail. Analysis for Failure case is in Appendix H These anecdotal examples illustrate plausible mechanisms; for definitive evidence, see Table 1.

# 6 CONCLUSION

We introduced a reinforcement learning framework that uses the Lean proof assistant as a process-level reward oracle. Unlike prior outcome-only methods, our approach leverages Lean's parsing and validation to provide both global outcome signals and fine-grained tactic rewards, integrated into a GRPO objective. This enables denser, verifiable credit assignment: outcome rewards enforce proof-level success, while tactic rewards guide step-level reasoning. Experiments on STP-Lean and DeepSeek-Prover-V1.5 show consistent improvements on MiniF2F and ProofNet, with stable gains achieved by assigning tactic rewards to the first token of each tactic and first error propagation method. Overall, proof assistants can serve not only as checkers at inference but also as structured feedback sources during training, pointing toward more stable and effective RL for reasoning.

## LIMITATIONS

We did not compare against learned PRMs, as they rely on natural-language CoT supervision and large annotated datasets that are not yet available for Lean. Our models also generate pure Lean proofs without long CoT, leaving open how to design fine-grained rewards for long-form reasoning. In addition, tactic rewards in our method were fixed scores $(d_1, d_2)$, which proved effective but somewhat sensitive across different models and datasets. Developing general advantage estimators and large-scale tactic-level datasets remains important future work.

## ETHICS STATEMENT

This research does not involve human subjects, personal data, or sensitive information that could raise ethical concerns. All experiments were conducted on publicly available formal mathematical datasets, and the proposed models are solely trained and evaluated for automated theorem proving tasks.

## REPRODUCIBILITY STATEMENT

To ensure the reproducibility of our work, we provide a comprehensive description of our and training process in Section 5.1. For further details such as hyperparameter, version of Lean, we introduced it in Appendix B. We utilized Huggingface and trl library for our experiments.

## ACKNOWLEDGEMENT

This work was supported by the Institute of Information & Communications Technology Planning & Evaluation (IITP) grant funded by the Korean government (MSIT) [No. RS-2022-II220311, Development of Goal-Oriented Reinforcement Learning Techniques for Contact-Rich Robotic Manipulation of Everyday Objects, 90%] and [No. 2019-0-00075, Artificial Intelligence Graduate School Program (KAIST), 10%].

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

APPENDIX

## A ADDITIONAL RELATED WORKS

**Automatic Theorem Proving** An automated theorem prover typically consists of two stages. The first is the process of translating mathematical statements written in natural language into formal statements. Wu et al. (2022) utilized large language models to translate mathematical questions into formal languages such as Isabelle and HOL. This process, known as autoformalization, is primarily used for constructing datasets intended for formal reasoning. Benchmarks or training datasets such as MiniF2F, LeanWorkbook, ProofNet, Deepseek-Prover have employed LLMs to translate natural language mathematical statements into formal expressions, contributing to the creation of high-quality formal reasoning datasets (Zheng et al., 2022; Azerbayev et al., 2023; Xin et al., 2024a; Ying et al., 2024a).

The second stage involves generating a formal proof from the translated formal statement. This proof generation process is typically divided into two approaches: one involves step-by-step inference, such as tree search during inference time (Polu & Sutskever, 2020; Azerbayev et al., 2024; Wu et al., 2024; Xin et al., 2024b), and the other generates the entire proof at once (Xin et al., 2024a; Lin et al., 2025b). Ospanov et al. (2025); Wang et al. (2025c); Baba et al. (2025) use Lean compiler as agent for complementing formal reasoning ability of LLMs, while (Dong & Ma, 2025) enhances formal reasoning by augmenting problems via conjecture. (Jiang et al., 2023) presents a unified framework that combines both autoformalization and proof generation in a single pipeline.

Existing methods such as Lean-STaR and RMaxTS (Lin et al., 2025a; Xin et al., 2024b) utilize Lean as a step-checker during inference, generating steps sequentially and searching optimally via tree search to find valid proofs. In contrast, in this paper, similar to (Wang et al., 2025a; Zhang et al., 2025; Ren et al., 2025; Ji et al., 2025), we utilize Lean as a whole-proof verifier during the training stage. Additionally, beyond merely providing correctness checks for the entire proof, we leverage Lean's parsing and elaboration capabilities to validate each individual tactic step, integrating this step-level validation into the training process. In other words, we employ the Lean proof assistant as a process-based reward model for validating the correctness of each reasoning steps. (Lightman et al., 2023).

Unlike prior work that leverages dense feedback from proof assistants, our research takes a different perspective: we rely solely on the rule-based signals of the symbolic engine, without introducing any natural language. Approaches such as Lin et al. (2025a;c); Wang et al. (2025b); Li et al. (2025),exploit natural language reasoning as a form of annotation to enhance LLMs formal reasoning abilities. In contrast, our method improves performance exclusively through reward signals provided by Lean, without any reliance on natural language.

**Reinforcement Learning in Language Models** While developing or applying algorithms such as PPO (Schulman et al., 2017) and GRPO (Shao et al., 2024) plays a significant role in reinforcement learning, reward shaping and credit assignment are central challenges in reinforcement learning. (Cobbe et al., 2021) introduced a reward model based on the outcome of a response. However, similar to other areas of RLHF, this approach suffers from the limitation of sparse rewards (Chan et al., 2024; Zheng et al., 2023).

To address this, process-based reward model (PRM) assigns step-level rewards during inference to guide rationale generation (Lightman et al., 2023), and can also be used to reward responses during training (Setlur et al., 2024; Kazemnejad et al., 2024). (Yuan et al., 2024; Cui et al., 2025) derived an implicit PRM from the ORM without any data annotation or additional training. When assigning scores to reasoning steps, (Lightman et al., 2023) defined the reward as the correctness of each step, which required substantial human annotation effort. (Wang et al., 2024) instead adopted a Monte Carlo approach, defining the score of a step as the proportion of successful rollouts originating from that step. While recent PRM approaches show promise in natural language reasoning, they require large annotated datasets of step-level correctness. To the best of our knowledge, no such dataset exists for Lean or formal theorem proving, making a direct comparison with a learned PRM baseline infeasible. This further motivates our approach of leveraging the Lean verifier itself as a process oracle. In contrast, our process-based reward leverages the Lean theorem prover to automatically

verify the correctness of each step, thereby eliminating the need for human annotators or sampling many proofs steps.

Our work can also be interpreted through the lens of reward shaping (Ng et al., 1999). Prior approaches have explored different mechanisms for distributing reward signals: Chan et al. (2024) leverages the internal attention patterns of LLMs to assign higher weights to important tokens, Cao et al. (2025) employs Shapley values to allocate credit across actions, and Kazemnejad et al. (2024) uses Monte Carlo rollouts to estimate and distribute rewards over intermediate steps. In contrast, our method relies on an external parser-the Lean theorem prover-to parse tactics and assign reward to the first token, thereby implementing a form of credit assignment.

## B  EXPERIMENTAL DETAIL

**Data.**  We randomly sampled 10k instances from the STP dataset (3.26M total) for RL training. For DeepSeek-Prover-V1.5-SFT, we applied an additional supervised fine-tuning step on 500k STP samples before RL, since the vanilla model produced low-quality proofs during RL training.

**Verification.**  We use Lean 4.9.0-rc1 for all experiments in the paper. During training, we used a REPL (read-eval-print loop) interface with Lean to verify proofs and assign outcome- and tactic-level rewards. Each proof attempt was given a maximum of 15 seconds for verification; longer runs were treated as failures (both outcomes, tactic rewards are zero).

**RL configuration.**  For GRPO training, we used $G = 4$ generations per prompt, sampling temperature 0.9, KL coefficient 0.04, clipping $\epsilon = 0.2$, and the DAPO upper bound 0.28 (Yu et al., 2025). Tactic-level rewards were fixed at $d_1 = -0.05$ and $d_2 = -0.1$ for partially valid and erroneous tactics in the main experiments, respectively. All experiments used non-CoT prompts, following Xin et al. (2024b).

**Training details.**  We fine-tuned the models with LoRA (rank 64, $\alpha = 64$) using bf16 precision. The AdamW optimizer was used with a learning rate of $1.0 \times 10^{-5}$. Maximum response length was set to 1024 tokens during both training and evaluation.

**Evaluation.**  For decoding we used temperature 1.0 and top-p 0.95. We re-evaluated all baselines under the same non-CoT and budget settings (32/64 samples). All reported results are from the final checkpoint.

**Compute.**  Training was conducted on $4 \times$ NVIDIA A6000 GPUs, requiring approximately 21-23 hours.

## C  HYPERPARAMETER ABLATIONS ON $d_i$

| Setting | Model Size | Sample Budget | MiniF2F - Test | ProofNet - Test |
|---|---|---|---|---|
| STP-baseline | 7B | 32 | $55.9\% \pm 0.2$ | $17.2\% \pm 0$ |
|  |  | 64 | $56.7\% \pm 0.2$ | $19.1\% \pm 0.4$ |
| GRPO baseline | 7B | 32 | $55.7\% \pm 1$ | $17.4\% \pm 0.6$ |
|  |  | 64 | $57.9\% \pm 0.5$ | $19\% \pm 0.3$ |
| $d_1 = -0.05, d_2 = -0.10$ | 7B | 32 | $57.1\% \pm 0.8$ | $18.6\% \pm 0.3$ |
|  |  | 64 | $59.2\% \pm 0.5$ | $19\% \pm 0.3$ |
| $d_1 = d_2 = -0.10$ | 7B | 32 | $57.7\% \pm 0.2$ | $17.6\% \pm 0.6$ |
|  |  | 64 | $58.7\% \pm 0.8$ | $18.1\% \pm 0.6$ |
| $d_1 = -0.05, d_2 = -0.50$ | 7B | 32 | $57\% \pm 0.4$ | $17.6\% \pm 0.3$ |
|  |  | 64 | $59.2\% \pm 0.5$ | $18.6\% \pm 0.8$ |

Table 5: Ablation study on tactic-level penalties $d_1, d_2$. We compare outcome-only GRPO baseline with three variants of $(d_1, d_2)$ settings. Results are reported as pass@32 and pass@64 (%) on MiniF2F and ProofNet test sets. The experiment is couducted with STP-Lean model.

This ablation shows that introducing a gap between $d_1$ and $d_2$ makes the method more robust: performance remains consistently above the GRPO baseline, with stable gains across different penalty scales and especially clear improvements on MiniF2F.

## D PROMPTS

For training and evaluation, we used non-COT evaluation followed by (Dong & Ma, 2025) and (Xin et al., 2024b). The examples are introduced in Table 6, 7.

---

**Prompt Template**

```
Complete the following Lean 4 code:\n\n
```lean4\n{header}{formal_statement}
```

---

Table 6: Prompt template used in training and evaluation. We selected non-COT generation which is appropriate with our MDP setting

---

**Prompt Example**

```
Complete the following Lean 4 code:

```lean4
import Mathlib
import Aesop
set_option maxHeartbeats 0
open BigOperators Real Nat Topology Rat

theorem theorem_exercise_2011_2_257 (G : Type*) [Group G] [Fintype
↪   G]
(h : Fintype.card G | 2) (x : G) : x ^ 2 = 1
  ( x y : G, x * y = y * x)   ( a : G, a = aź)   a : G, a^2 = 1
  let p x : G  G := by
```

---

Table 7: A training sample used in training and evaluation. We selected non-COT generation which is appropriate with our MDP setting.

## E CREDIT ASSIGNMENT IN REINFORCEMENT LEARNING

Let $y_t$ be the $t$-th token of $y$, $R$ denote the reward model, $\pi_\theta$ represent the policy model, and $\pi_{\mathrm{ref}}$ be the reference model. $L$ denote the response length and $B$ be a coefficient controlling the distance between the policy and the reference policy. In PPO, the token-level reward at position $t$ is defined as: $r_t(x, y_t) = R(x, y)\mathbf{1}(y_t = L) - B\log\left(\frac{\pi_\theta(y_t|x)}{\pi_{\mathrm{ref}}(y_t|x)}\right)$, where the non-zero reward $R(x, y)$ is assigned only to the last token. For all other tokens, only a KL divergence penalty is applied via a log ratio $\log\left(\frac{\pi_\theta(y_t|x)}{\pi_{\mathrm{ref}}(y_t|x)}\right)$. Direct usage of rewards can lead to high variance; therefore, PPO reduces variance by utilizing a learned value model $V$. This value network assigns a value to each token $y_t$, from which the Temporal-Difference (TD) error is computed as: $\delta_t = r_t + \gamma V(y_{t+1}) - V(y_t)$ where $\gamma$ is discounted factor. Then, the advantage for each token is recursively calculated as follows: $A_L = \delta_L, A_t = \delta_t + \gamma\lambda A_{t+1}$, for $t = L - 1, L - 2, \ldots, 1$. Subsequently, because the computed advantages $A_t$ can exhibit high variance during exploration, normalization or similar techniques are applied, resulting in the final adjusted advantage $A_t$. This adjusted advantage is then utilized in the PPO loss defined as:

$$L^{\mathrm{CLIP}}(\theta) = \mathbb{E}_t\left[\min\left(\frac{\pi_\theta(y_t \mid x)}{\pi_{\theta_{\mathrm{old}}}(y_t \mid x)}A_t,\ \mathrm{clip}\left(\frac{\pi_\theta(y_t \mid x)}{\pi_{\theta_{\mathrm{old}}}(y_t \mid x)},\ 1 - \epsilon,\ 1 + \epsilon\right)A_t\right)\right] \qquad (2)$$

In contrast, REINFORCE-based methods such as GRPO and RLOO have proposed algorithms that optimize policies directly from verifiable rewards without requiring a value model, due to concerns about the computational cost and estimation capability associated with training value networks.

GRPO generates multiple response groups $\{ y_{(i)} \}_{i=1}^{G}$ for a given question $q$ from an old policy $\pi_{\mathrm{old}}$. Subsequently, a reward function outputs reward $r = \{ r_{(i)} \}_{i=1}^{G}$ for each response group. If we set $y_{i,t}$ as $t-$th token index of response $y_i$ The advantage for $y_{i,t}$, $A_{i,t}$ is then computed by normalizing these rewards as follows: $\hat{A}_{i,t} = \frac{r_i - \mathrm{mean}(r)}{\mathrm{std}(r)}$.

This advantage is uniformly assigned to each token $y_{i,t}$ constituting the response $y_i$. Subsequently, this identical token-level advantage is utilized in calculating the following loss:

$$L_{\mathrm{GRPO}}(\theta) = \mathbb{E}_{q \sim P(Q),\, \{ y_i \}_{i=1}^{G} \sim \pi_{\theta_{\mathrm{old}}}(Y|q)}$$
$$\left[ \frac{1}{G} \sum_{i=1}^{G} \Big\{ \min\Big( \frac{\pi_\theta(y_{i,t} \mid q)}{\pi_{\theta_{\mathrm{old}}}(y_{i,t} \mid q)}\, \hat{A}_{i,t}, \mathrm{clip}\Big( \frac{\pi_\theta(y_{i,t} \mid q)}{\pi_{\theta_{\mathrm{old}}}(y_{i,t} \mid q)}, 1 - \epsilon,\, 1 + \epsilon \Big)\, \hat{A}_{i,t} \Big) \,-\, \beta\, D_{\mathrm{KL}}\big[ \pi_\theta \,\|\, \pi_{\mathrm{ref}} \big] \Big\} \right]. \quad (3)$$

As our goal is to integrate Lean-based fine-grained rewards to GRPO algorithm, our method keep balance between value-based PPO's fine-grained credit assignmethod and GRPO's simple unifrom credit assignment approach.

## F    GENERATED PROOFS EXAMPLES

**imo_1960_p2 in minif2f from tactic-reward-trained model**

```
import Mathlib
import Aesop

set_option maxHeartbeats 0

open BigOperators Real Nat Topology Rat

theorem imo_1960_p2 (x : \mathbb{R}) (h_0 : 0  1 + 2 * x)
(h_1 : (1 - Real.sqrt (1 + 2 * x)) ^ 2  0)
    (h_2 : 4 * x ^ 2 / (1 - Real.sqrt (1 + 2 * x)) ^ 2 < 2 * x + 9) :
    -(1 / 2)  x  x < 45 / 8 := by
  have h_3 : 0  (1 + 2 * x) := by linarith
  have h_4 : 0  Real.sqrt (1 + 2 * x):= Real.sqrt_nonneg(1 + 2 * x)
  have h_5 : 0  (1 + 2 * x) * Real.sqrt (1 + 2 * x) := by positivity
  have h_6 : 0 < (1 - Real.sqrt (1 + 2 * x)) ^ 2 := by
    apply sq_pos_of_ne_zero
    intro h
    apply h_1
    nlinarith
  constructor
  nlinarith [sq_sqrt (show 0  1 + 2 * x from by linarith), h_2]
  have h_7 : x < 45 / 8 := by
    apply lt_of_not_ge
    intro hge
    have h_8 : 4 * x ^ 2 / (1 - Real.sqrt (1 + 2 * x)) ^ 2  2*x+9
    := by
      rw [ge_iff_le]
      rw [le_div_iff h_6]
      nlinarith [sq_sqrt (show 0  1 + 2 * x from by linarith), hge]
    linarith
  linarith
```

Table 8: Proof of `imo_1960_p2` from MiniF2F generated by our tactic-reward-trained model.

**imo_1960_p2 in minif2f from baseline model**

```
import Mathlib
import Aesop

set_option maxHeartbeats 0

open BigOperators Real Nat Topology Rat

theorem imo_1960_p2 (x : \mathbb{R})
(h_0 : 0  1 + 2 * x) (h_1 : (1 - Real.sqrt (1 + 2 * x)) ^ 2  0)
    (h_2 : 4 * x ^ 2 / (1 - Real.sqrt (1 + 2 * x)) ^ 2 < 2 * x + 9) :
    -(1 / 2)  x  x < 45 / 8 := by
  have h_3 : 0 < (1 - Real.sqrt (1 + 2 * x)) ^ 2 := by positivity
  have h_4 : 0  Real.sqrt (1 + 2 * x) := Real.sqrt_nonneg _
  have h_5 : 0  1 + 2 * x := by linarith
  have h_6 : Real.sqrt (1 + 2 * x)  1 + x := by
    rw [Real.sqrt_le_iff]
    constructor
    ů linarith
    ů nlinarith
  have h_7 : 4 * x ^ 2 / (1 - Real.sqrt (1 + 2 * x)) ^ 2 < 2 * x + 9
  :=
  by assumption
  have h_8 : -(1 / 2)  x := by
    rw [div_lt_iff h_3] at h_7
    nlinarith
  have h_9 : x < 45 / 8 := by
    rw [div_lt_iff h_3] at h_7
    nlinarith
  exact h_8, h_9
  <;> assumption
  <;> assumption
  <;> assumption
  <;> assumption
  <;> assumption
  <;> assumption
  <;> assumption
  <;> assumption
  <;> assumption
```

Table 9: Proof of `imo_1960_p2` from MiniF2F generated by the baseline STP model.

**exercise_1_14 in ProofNet from tactic-reward-trained model.**

```
import Mathlib

open Topology Filter Real Complex TopologicalSpace Finset
open scoped BigOperators

theorem exercise_1_14
  (z : \mathbb{C}) (h : abs z = 1)
  : (abs (1 + z)) ^ 2 + (abs (1 - z)) ^ 2 = 4 := by
  rw [Complex.normSq_eq_abs, Complex.normSq_eq_abs]
  simp [Complex.normSq_add, Complex.normSq_sub, h]
  ring_nf
  <;> simp [h, Complex.normSq_eq_abs]
  <;> ring_nf
```

Table 10: Proof of `exercise_1_14` from ProofNet generated by our tactic-reward-trained model.

**exercise_1_14 in ProofNet from baseline**

```
import Mathlib

open Topology Filter Real Complex TopologicalSpace Finset
open scoped BigOperators

theorem exercise_1_14
  (z : \mathbb{C}) (h : abs z = 1)
  : (abs (1 + z)) ^ 2 + (abs (1 - z)) ^ 2 = 4 := by
  have h_0 : abs z ^ 2 = 1 := by simp [h]
  simp [pow_two, abs_add, abs_sub, h_0]
  ring_nf
  simp_all [sq_abs]
```

Table 11: Proof of `exercise_1_14` from ProofNet generated by the baseline STP model.

# G  RESULTS ON RETURN-BASED ADVANTAGE

| Model | Model Size | Sample Budget | MiniF2F - Test | ProofNet - Test |
|---|---|---|---|---|
| Return | 7B | 32 | $55.3\% \pm 0.4$ | $18.1\% \pm 0.3$ |
| | | 64 | $57.5\% \pm 0.2$ | $18.6\% \pm 0.3$ |
| Outcome+Tactic RL (ours) | 7B | 32 | $\mathbf{57.1\%} \pm 0.8$ | $\mathbf{18.6\%} \pm 0.3$ |
| | | 64 | $\mathbf{59.2\%} \pm 0.5$ | $\mathbf{19\%} \pm 0.3$ |

Table 12: Ablation study on reward strategies for return-based advantage.

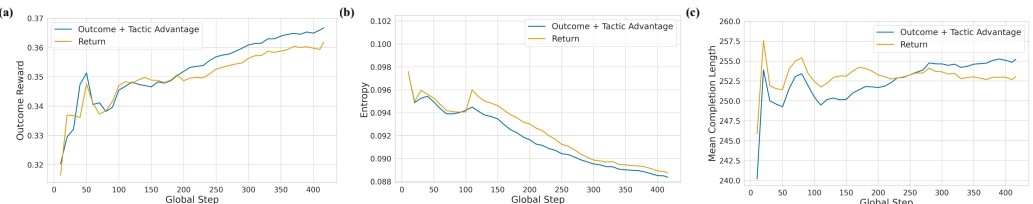

Figure 4: Training dynamics showing (a) outcome reward,(b) entropy, and (c) mean of response length during reinforcement learning in our method and return-based method.

## H   FAILURE CASE AND LIMITATION OF OUR METHOD

**imo_2019_p1 in MiniF2f from tactic-reward-trained model.**

```
import Mathlib
import Aesop

set_option maxHeartbeats 0

open BigOperators Real Nat Topology Rat

theorem imo_2019_p1 (f : \Z   \Z) :
    (\forall a b, f (2 * a) + 2 * f b = f (f (a + b)))
    \n \iff (\forall z, f z = 0)
    \n (\exists c, \forall z, f z = 2 * z + c) := by
  constructor
  intro h
  intro z
  have h_1 := h 0 0
  have h_2 := h 0 1
  have h_3 := h 1 0
  have h_4 := h 1 1
  have h_5 := h 2 0
  have h_6 := h 2 1
  have h_7 := h 0 2
  have h_8 := h 1 2
  have h_9 := h 2 2
  simp at h_1 h_2 h_3 h_4 h_5 h_6 h_7 h_8 h_9
  norm_num at h_1 h_2 h_3 h_4 h_5 h_6 h_7 h_8 h_9
  omega
  intro h
  intro a b
  have h_1 := h (2 * a)
  have h_2 := h b
  have h_3 := h (a + b)
  cases' h_1 with h_1 h_1 <;> cases' h_2 with h_2 h_2
  \n <;> cases' h_3 with h_3 h_3 <;> simp_all
  <;> omega
```

Table 13: Proof of `imo_2019_p1` in MiniF2f generated by our tactic-reward-trained model.

Consider a function $f : \mathbb{Z} \to \mathbb{Z}$ satisfying

$$\forall a, b \in \mathbb{Z}, \qquad f(2a) + 2\,f(b) \;=\; f\big(f(a+b)\big).$$

The task is to prove that necessarily one of the following holds:

   (i) $\forall z \in \mathbb{Z},\ f(z) = 0$, or
   (ii) $\exists c \in \mathbb{Z},\ \forall z \in \mathbb{Z},\ f(z) = 2z + c$.

Our model first introduced the assumption

$$h :\ \forall a, b \in \mathbb{Z},\ f(2a) + 2\,f(b) = f\big(f(a+b)\big),$$

and then instantiated it at several concrete pairs to create hypotheses $h_i$ (e.g., $h_1 := h(0,0)$, $h_2 := h(0,1)$, …). After some local simplification steps (e.g., `simp`, `norm_num`), it attempted to close the goal using the `omega` tactic, a decision procedure for Presburger arithmetic (linear integer arithmetic).

However, the `omega` call produced the first Lean error. While our method correctly assigns the $d_2$ penalty to this failing `omega` tactic under first-error propagation, it does not penalize the preceding tactics (`intro`, `have`, `simp`) because they elaborate successfully and thus appear locally valid. In other words, although introducing $h$ and instantiating $h_i$ is not logically incorrect, this route is strategically unproductive for this problem: the remaining goal still involves quantifiers, disjunction, and

an uninterpreted function $f$, which lie outside `omega`'s theory. Consequently, our current scheme only punishes the terminal failing step and fails to capture that the earlier (locally successful) steps did not make meaningful progress toward solving the global goal.

## I   LARGE LANGUAGE MODEL USAGE

In preparing this manuscript, we made limited use of large language models strictly for writing assistance. Specifically, we used ChatGPT-5 and Gemini-2.5 to improve grammar, enhance clarity of expression, and polish the overall presentation.

## J   MATHEMATICAL GROUND FOR TACTIC REWARD

In this section, we provide a conceptual interpretation of our tactic-level rewards using a simple value model and potential-based reward shaping. Our goal is not to claim a formal optimality guarantee, but rather to clarify how the structure of our discrete Lean-based rewards is aligned with an idealized concept of proof success under a First error propagation assumption with potential function.

### J.1   DEFINE VALUE FUNCTION

Consider a Lean proof trajectory

$$s_0 \xrightarrow{T_1} s_1 \xrightarrow{T_2} \cdots \xrightarrow{T_N} s_N,$$

where $s_t$ denotes the prefix of tactics $(T_1, \ldots, T_{t-1})$, and $s_N$ refers to a completed proof.

We adopt the modeling assumption already used in the main paper:

From the first error propagation, once the first erroneous tactic occurs, the proof can no longer be repaired into a valid Lean proof.

Formally, let $j$ be the index of the first tactic for which Lean reports an error. Then all states $s_t$ with $t \geq j$ lie in an absorbing failed state. Instead of assuming independent Bernoulli errors, we consider a more general and realistic model with conditional valid probabilities. For a valid prefix $s_{k-1}$, let

$$q(s_{k-1}) = P(\text{no error at step } k \mid s_{k-1} \text{ is valid}).$$

Under the first error propagation assumption, we define the value function as the probability of eventually producing a valid proof from a valid prefix $s_t$ is

$$V(s_t) = P(\text{success} \mid s_t) = \prod_{k>t} q(s_{k-1}).$$

Along a valid trajectory, we have

$$V(s_{t+1}) = \frac{V(s_t)}{q(s_t)} \geq V(s_t),$$

so $V(s_t)$ is monotone increasing until no errors are founded. if the first error occurs at step $j$, the success probability collapses to zero:

$$V(s_j) = V(s_{j+1}) = \cdots = 0.$$

Qualitatively, this yields the following structure:

- For a successful proof, $V(s_t)$ increases from a small value at $s_0$ to $V(s_N) = 1$.
- For a failed proof, $V(s_t)$ increases along the correct prefix, and then drops to 0 at the first error and stays at 0 afterwards.

Thus, the ideal value function encodes (i) Monotone growth along error-free prefixes and (ii) Irreversible collapse after the first error.

This structure motivates using stronger positive credit for tactics on an error-free prefix and negative or neutral credit after the first failure.

## J.2 POTENTIAL-BASED REWARD SHAPING WITH $V(s)$

The value defined in previous section suggests a way to define potential-based shaping. In an idealized MDP setting where the environment state is exactly $s_t$ and the agent has access to the value function $V(s)$, one could define a potential

$$\Phi^\star(s) = f\big(V(s)\big),$$

where $f$ is any monotonically increasing transformation (e.g., $f(v) = v$). A shaped reward can then be written as

$$r_t^\star = r_{\text{outcome},t} + \gamma\Phi^\star(s_{t+1}) - \Phi^\star(s_t),$$

where $r_{\text{outcome},t}$ is the sparse end-of-proof reward derived from $g(Y)$. Under the assumptions of (Ng et al., 1999), such potential-based shaping preserves the set of optimal policies.

Intuitively, using $\Phi^\star(s) = f(V(s))$ means that the potential is highest on globally successful trajectories, increases along error-free prefixes, and collapses after the first error, similar with the structure of $V(s)$ in previous section. The corresponding temporal-difference term

$$\gamma\Phi^\star(s_{t+1}) - \Phi^\star(s_t)$$

acts as a local improvement signal: it is positive along valid contexts, negative when the value collapses at the first error, and zero afterwards.

In our setting, however, we neither assume access to the true $V(s)$. We therefore view this potential-based construction as a *normative model* that suggests the qualitative shape of a desirable local credit signal, rather than as a source of formal optimality guarantees.

## J.3 LEAN-BASED DISCRETE APPROXIMATION AS QUANTIZED LOCAL SHAPING

In practice, we did not estimate $V(s)$ or $\Phi^\star(s)$ explicitly. Instead, we exploit Lean's symbolic feedback (AST errors, first-error propagation) to construct discrete tactic-level scores.

For a proof $Y$, with first error index $j$ (if any), recall that we define

$$\varphi(Y, T_t) = \begin{cases} 1, & \text{if } g(Y) = 1, \\ d_1, & \text{if } g(Y) = 0 \text{ and } t < j, \\ d_2, & \text{if } g(Y) = 0 \text{ and } t \geq j, \end{cases} \qquad 1 > d_1 > d_2.$$

as the process-level reward for tactic $T_t$.

Conceptually, $\varphi(Y, T_t)$ is a coarse, Lean-driven *quantization* of the ideal local improvement signal suggested by the value model. States on globally successful trajectories receive the highest score (1); states on error-free prefixes of failed proofs receive an intermediate score ($d_1$); and states at or after the first error receive the lowest score ($d_2$). This partitions trajectories into three regions whose ordering (success > pre-error > post-error) is aligned with the ordering of $V(s)$ implied by previous section.

Assuming $\gamma = 1$ and a finite horizon, any such per-step process reward sequence can be written as a difference of a state potential. For a fixed trajectory $Y$ of length $N$, define $\Phi$ backwards by

$$\Phi(s_N) = 0, \qquad \Phi(s_{t+1}) - \Phi(s_t) = r_{\text{process},t}, \quad t = N - 1, \ldots, 0.$$

By construction,

so the total shaped reward becomes

$$r_t' = r_{\text{outcome},t} + r_{\text{process},t} = r_{\text{outcome},t} + \Phi(s_{t+1}) - \Phi(s_t).$$

This potential $\Phi$ is not intended as an estimate of the true value function $V(s)$; rather, it is an implicit potential induced by our discrete Lean-based scoring rule. The key point is that its level sets respect the same qualitative ordering (success > pre-error > post-error) as the ideal value model, providing a theoretically motivated yet practical shaping signal.

### J.4 DISCUSSION AND LIMITATIONS

Our analysis suggests the following:

- Under a first-error propagation assumption, an ideal value function $V(s)$ for Lean proofs increases along error-free prefixes and collapses to zero after the first error.
- Our discrete tactic-level scores $\varphi(Y, T_t) \in 1, d_1, d_2$ can be viewed as a quantized local improvement signal, capturing this qualitative structure without estimating $V(s)$ explicitly.
- For any given trajectory, the resulting process rewards $r_{\text{process},t}$ can be written as a potential-based shaping term $r_{\text{process},t} = \Phi(s_{t+1}) - \Phi(s_t)$ for a suitable potential $\Phi$.

Consequently, our use of potential-based shaping should be understood as a theoretical framework that explains the structure of our rewards and motivates our design choices, rather than as a strict proof that our procedure preserves the optimal policy for the original sparse outcome reward. Empirically, we observe that this verifier-informed, discretized shaping leads to more stable training and consistent improvements over outcome-only GRPO on MiniF2F and ProofNet. We emphasize that we do not claim any formal optimality guarantee for this shaped reward in our large-scale LLM and Lean setting; the potential-based perspective is used purely as a conceptual framework for designing and interpreting our tactic-level rewards.

