# OpenReview forum: "Process-Verified Reinforcement Learning for Theorem Proving via Lean"
_ICLR.cc/2026/Conference — ICLR 2026 Poster_

### Official Review · Reviewer_sb7Y · 2025-10-31

**Soundness:** 2
**Presentation:** 3
**Contribution:** 1
**Rating:** 2
**Confidence:** 4

**Summary:**

The paper uses Lean as a verifier for process oracle rewards (which are tactic-level rewards in the case of theorem proving). It then incorporates it into a GRPO-style reinforcement learning objective with first-error propagation and first-token credit methods that balances outcome- and process-level advantages. The authors report that on MiniF2F and ProofNet, the proposed approach outperforms both outcome-only RL and vanilla baselines.

**Strengths:**

- The paper is well-written and easy to understand. This includes a dedicated formalization of the problem and clear mathematical representation.
- The approach is straightforward. Once the additional process rewards are defined, incorporating them with GRPO seems standard and I believe reproducible.

**Weaknesses:**

- My major concern is novelty. Leveraging feedback from ITPs has long been known to (and practiced by) the ML for theorem proving community. The paper doesn’t offer much other than demonstrating that combining it with GRPO did bring some benefits empirically. This is particularly concerning because the tactic-level rewards being considered here essentially only distinguish between “applicable” and “inapplicable” tactic applications. While useful, they are not novel signals.

**Questions:**

Have you thought about other more fine-grained tactic-level rewards?

---

> ### Author Response · Authors · 2025-11-19
>
> ### **Q1) Novelty of using Lean feedback**
> We appreciate the reviewer’s perspective and agree that using proof assistant feedback is widely practiced in ML for theorem proving. Our goal is not to claim that “Applying ITP feedback” itself is new, but rather to instantiate and analyze it as a process-level reward oracle within a GRPO-style RL framework.
>
> That is, beyond simply using ITP feedback, our aim is to **formalize and study how useful symbolic fine-grained signals can be delivered to an LLM in a way that is actually effective for training**.
>
> While applicable/inapplicable may look simple, the key challenge is that Lean outputs symbolic, tactic-structured language information, whereas language models operate on autoregressive token sequences and using score-level reward during RL training.
>
> The main novelty of our work is showing
>
> ***How to bridge this mismatch through a concrete credit assignment mechanism:*** tactic-level MDP, first-error propagation, and token-level credit assignment.
>
> Bridging this representational mismatch is nontrivial and has not been systematically addressed in prior work, and our ablations demonstrate that these design choices materially impact stability and performance.
>
> We clarify our contributions as follows, in comparison with past work:
>
> **Process oracle formulation & Credit assignment for GRPO integration.**
>
> Prior work typically uses Lean in one of two ways:
>
> (i) as a *step checker* at inference time to drive tree search [1,2] or
>
> (ii) as a *whole-proof verifier* to give binary feedback to generated proofs for training [2,3].
>
> Although [4] attempted to exploit fine-grained tactic-level signals in an online RL algorithm in their Appendix, it did not show formalization or performance increase.
>
> In contrast, we **treat Lean as a process oracle that returns both a global outcome and fine-grained tactic-level scores for every trajectory**, and we **integrate these feedback signals directly into a GRPO-style objective at the token level**.
>
> This requires several concrete design steps:
>
> 1. A formulation for integrating symbolic fine-grained signals into our training algorithm.
>     : We formalized a tactic-level MDP
> 2. A tactic-level score shaping strategy aligned with the autoregressive nature of LLM reasoning
>     : We chose first-error propagation rule
> 3. Mapping the per-tactic scores defined in step 2 into token-level advantages.
>     : We designed first-token assignment
>
> To our knowledge, ours is the first work to *systematically construct and ablate* this credit-assignment pipeline for online RL in theorem proving. Section 5.3 demonstrates that these design choices materially affect stability and performance, whereas prior work [4] attempted to use fine-grained signals but did not formalize the mechanism or obtain measurable benefit.
>
> **Furthermore, to clarify the mathematical grounding of our reward design, we added a new Appendix section (Appendix J) that provides a theoretical interpretation of our tactic-level rewards. We show that (i) an idealized first error propagation rule induces a monotone value structure, and (ii) our discrete Lean-based scores can be viewed as a coarse potential-based shaping term consistent with this structure.**
>
> ### **Q2) On more fine-grained tactic-level rewards**
> We also attempted several additional approaches to utilize richer ITP information.
>
> For example, we experimented with using Lean’s parser information (e.g., the distance to the first error within a tactic), or defining reward returns more closely aligned with standard RL formulations. These did not provide additional benefits within our compute budget.
>
> We found that our simple but structured scheme—combined with first-error propagation and first-token assignment—already provides **consistent improvements over both the baseline and outcome-only GRPO** on two benchmarks.
>
> We agree that more fine-grained signals could further improve performance, and our framework is designed to support such extensions such as goal-based reward.

---

> ### Author Response · Authors · 2025-11-19
>
> We sincerely thank the reviewer for raising these points. We will revise the paper to make our contributions clearer and more aligned with the reviewer’s suggestions. We appreciate the thoughtful feedback and would be glad to address any follow-up questions or continue the discussion.
>
> [1] Haohan Lin, Zhiqing Sun, Sean Welleck, and Yiming Yang. Lean-star: Learning to interleave thinking and proving, 2025
>
> [2]Huajian Xin, Z. Z. Ren, Junxiao Song, Zhihong Shao, Wanjia Zhao, Haocheng Wang, Bo Liu, Liyue Zhang, Xuan Lu, Qiushi Du, Wenjun Gao, Qihao Zhu, Dejian Yang, Zhibin Gou, Z. F. Wu, Fuli Luo, and Chong Ruan. Deepseek-prover-v1.5: Harnessing proof assistant feedback for reinforcement learning and monte-carlo tree search, 2024
>
> [3] Haiming Wang, Mert Unsal, Xiaohan Lin, Mantas Baksys, Junqi Liu, Marco Dos Santos, Flood Sung, Marina Vinyes, Zhenzhe Ying, Zekai Zhu, Jianqiao Lu, Hugues de Saxcé, Bolton Bailey, Chendong Song, Chenjun Xiao, Dehao Zhang, Ebony Zhang, Frederick Pu, Han Zhu, Jiawei Liu, Jonas Bayer, Julien Michel, Longhui Yu, Léo Dreyfus-Schmidt, Lewis Tunstall, Luigi Pagani, Moreira Machado, Pauline Bourigault, Ran Wang, Stanislas Polu, Thibaut Barroyer, Wen-Ding Li, Yazhe Niu, Yann Fleureau, Yangyang Hu, Zhouliang Yu, Zihan Wang, Zhilin Yang, Zhengying Liu, and Jia Li. Kimina-prover preview: Towards large formal reasoning models with reinforce- ment learning, 2025
>
> [4]Xingguang Ji, Yahui Liu, Qi Wang, Jingyuan Zhang, Yang Yue, Rui Shi, Chenxi Sun, Fuzheng Zhang, Guorui Zhou, Kun Gai: Leanabell-Prover-V2: Verifier-integrated Reasoning for Formal Theorem Proving via Reinforcement Learning 2025

---

> ### Author Response · Authors · 2025-11-28
> **Gentle reminder**
>
> Dear Reviewer sb7Y,
>
> Thank you again for your thoughtful and constructive feedback on our work.
> We have now posted our detailed response to your review. If you have a moment, we would greatly appreciate it if you could take a look, as we aimed to address each of your concerns as clearly as possible. If anything remains unclear or if further discussion would be helpful, please feel free to let us know.
>
>
> We sincerely appreciate your time during this busy review period.

---

> > ### Comment · Reviewer_sb7Y · 2025-11-28
> >
> > Dear Authors,
> >
> > Thanks for the response and the reminder. The response did help me better understand the goal of the submission. I am happy to raise my score assuming the authors also include a clarification in the revision.

---

> ### Author Response · Authors · 2025-11-28
>
> Dear Reviewer sb7Y,
>
> Thank you for your thoughtful follow-up and for letting us know that the clarification was helpful.
> Following your suggestion, we have added clearer explanations in the revised manuscript:
>
> **Lines 69–79**: We provide a more explicit description of the goal and the motivation behind our formulation.
>
> **Lines 86–91**: We refine the presentation of our contribution, clarifying the structure and role of our credit-assignment framework.
>
> **Lines 106–107**: In the Related Work section, we more clearly distinguish our approach from previous methods and highlight our setting.
>
> Please feel free to let us know if any further clarification would be helpful. We sincerely appreciate your constructive feedback.

---

### Official Review · Reviewer_4VNo · 2025-10-31

**Soundness:** 3
**Presentation:** 2
**Contribution:** 2
**Rating:** 4
**Confidence:** 3

**Summary:**

This paper proposes using the Lean proof assistant as a symbolic process oracle during reinforcement learning training for theorem proving. The key idea is to parse generated proofs into tactic sequences and leverage Lean's elaboration feedback to assign tactic-level rewards in addition to outcome-level rewards. The authors integrate these structured rewards into a GRPO-style objective with first-error propagation and first-token credit assignment. Experiments on STP-Lean and DeepSeek-Prover-V1.5 show modest improvements on MiniF2F and ProofNet benchmarks over outcome-only baselines.

**Strengths:**

1. Clear formulation and motivation: The paper provides a well-structured formalization of Lean as a process oracle, defining parsing function f, outcome function g, and tactic-level scoring function φ. The MDP formulation is clear and the integration with GRPO is technically sound.

2. Comprehensive ablation studies: The authors conduct thorough ablations on token-level credit assignment (Table 3), reward strategies (Table 4), timeout settings (Figure 3), and hyperparameter sensitivity (Table 5, Appendix C), demonstrating careful experimental work.

3. Stability analysis: The comparison with outcome-only GRPO shows that tactic-level supervision provides more stable training dynamics, with analysis of entropy and proof length providing useful insights.

**Weaknesses:**

1. **Fundamental semantic limitation without learned PRM comparison (Critical)**: The paper's most significant limitation is that Lean's tactic-level feedback only provides **syntactic/type-theoretic correctness**, not **mathematical progress or strategic value**. As the authors acknowledge in Appendix H (imo_2019_p1 example), tactics that elaborate successfully (e.g., introducing hypotheses h_1 through h_9) receive no penalty despite being mathematically unproductive. The method only penalizes the final failing tactic (omega), missing that the entire proof strategy was flawed from the beginning.

   More critically, the paper does not compare against learned process reward models (e.g., Lightman et al., Wang et al.) that could potentially capture mathematical progress beyond syntactic verification. While the authors claim no Lean tactic-level dataset exists, this actually highlights the core limitation—**symbolic verification captures local soundness but misses global strategic value that learned PRMs aim to model**. The absence of this comparison makes it impossible to assess whether the symbolic approach is fundamentally limited compared to learned alternatives, or whether combining both approaches could address the semantic gap.

2. **Marginal empirical improvements**: The performance gains are quite limited:
   - MiniF2F: +1.2-1.4% at pass@32, +2.5% at pass@64
   - ProofNet: +1.4% at pass@32, essentially no gain (-0.1%) at pass@64

   These improvements are within typical variance ranges and raise questions about whether the added complexity is worthwhile. The gains are particularly underwhelming considering that the method requires calling Lean's parser and elaborator during training.

**Questions:**

The main questions and concerns are detailed in the Weaknesses section above.

---

> ### Author Response · Authors · 2025-11-19
>
> We appreciate the reviewer's insightful feedback on this critical issue. We fully agree that Lean’s tactic-level feedback only guarantees **local type-theoretic-based logicality or soundness**, and does not directly measure “mathematical progress” or global strategic value. We would like to highlight that our goal in this work is not to claim that symbolic verification replaces trained process reward models (PRMs), but rather to study how far one can go using **only the existing Lean verifier as a process oracle**, without any additional learned reward model or step-level annotations.
>
> ### **Q1)Limitation of symbolic verifier**
>
> We believe that even such *local*, logic-based symbolic verification remains valuable. Existing PRM-based or LLM-based verifiers ([1], [2]) still struggle with reliably checking logical validity, and LLMs often generate statements that are not logically well-formed. In particular, PRMs trained via Monte Carlo sampling to estimate the *future return* of each step, rather than explicitly and finely checking the logical consistency between steps. That is, PRMs predict how likely a partial reasoning trajectory is to eventually produce a correct final answer.
>
> Hence, we view symbolic Lean feedback and learned PRMs as **complementary**. PRMs are designed to capture semantic mathematical progress from noisy human/LLM supervision in natural language. In contrast, our reward signal is weaker semantically but **exactly sound by construction** under dependent type theory and automatically available for any Lean proof without extra data collection. The contribution of this paper is to show that **even this limited, purely symbolic signal**, when mapped carefully to tactics and tokens via first-error propagation and difficulty-normalized baselines, already yields **consistent improvements over outcome-only GRPO** on MiniF2F and ProofNet, across two different prover models.
>
> Our method does not claim to distinguish all “mathematically uninformative” prefixes from all other failures; instead, it uses Lean to separate *locally invalid* steps from *locally valid* ones, and empirically this is already sufficient to stabilize optimization relative to outcome-only RL. We agree that capturing semantic usefulness (e.g., mathematical informativeness, well-directed tactics) is an important open problem, and we will make this limitation more explicit in the final paper.
>
> ### **Q2)Comparison between trained PRM**
>
> We appreciate this concern and agree that a full comparison with a learned PRM would indeed help clarify whether symbolic feedback alone is sufficient or whether combining the two closes the semantic gap. However, integrating existing PRMs into the Lean setting is **not** straightforward: current PRMs are trained exclusively on *natural-language* CoT traces, not on Lean tactics, and cannot directly evaluate or score formal proof steps. Making such a comparison would require designing a new interface or collecting a large tactic-level PRM dataset, an interesting direction, but beyond the scope of this work.
>
> Our aim in this paper is therefore not to determine whether symbolic feedback outperforms PRMs, but rather to characterize what can be achieved in the “zero-annotation, symbolic-only” regime, where Lean is the only process oracle available. **We believe this is a meaningful setting because symbolic verification provides exact type-theoretic soundness, which current PRMs and LLM-based verifiers still struggle to guarantee**. **Our results show that even without PRMs, Lean-derived tactic signals already provide *stable gains* over outcome-only RL.**
>
> We agree that combining symbolic verification with a learned PRM is a promising next step to address the semantic gap, and we will make this future direction explicit in the revised paper.

---

> > ### Author Response · Authors · 2025-11-19
> >
> > ### **Q3) Magnitude of empirical improvements and cost.**
> >
> > We appreciate the reviewer’s concern about effect size. We would like to emphasize two points:
> >
> > (1) **Low additional complexity.**
> >
> > Our RL loop already calls Lean as a whole-proof verifier to compute outcome rewards, as in prior work. The proposed method reuses this same REPL interaction and adds only lightweight processing of Lean’s info trees (sorting tactics and applying first-error propagation).**Crucially, Lean must call its parser and elaborator even to compute the binary outcome reward**, because these components are required to interpret the entire proof, resolve tactic applications, and construct the abstract syntax tree before the kernel can check validity. As a result, **all tactic-level information we use—info trees, subgoal updates, and error logs—is already produced automatically as part of this standard verification pass.** We do not run any additional parsing or elaboration steps for our process-based rewards. Therefore, the only extra work needed is to **extract and reduce the already-generated information** down to token-level signals; no additional Lean verification passes or computation are required.
> >
> > (2) **Stability rather than sheer magnitude.** A central observation of our experiments is that **outcome-only GRPO can be unstable**, sometimes even underperforming the SFT baseline on ProofNet. Incorporating Lean tactic rewards yields **more stable improvements**: across both datasets and both base models, tactic-based training consistently matches or surpasses the supervised baseline and outcome-only GRPO, while keeping training cost nearly unchanged. While we acknowledge that the absolute performance gains are modest, we want to emphasize that they are **consistently positive and more stable** than those obtained with outcome-only RL.
> >
> > We thank the reviewer again for highlighting these points. We will revise the paper to make our scope explicit: symbolic Lean feedback is not intended to replace learned PRMs, but to establish a minimal, annotation-free baseline for tactic-level credit assignment. We view PRM integration—potentially using Lean feedback as supervision—as an important and complementary direction for future work.
> >
> > [1] Zhenru Zhang, Chujie Zheng, Yangzhen Wu, Beichen Zhang, Runji Lin, Bowen Yu, Dayiheng Liu, Jingren Zhou, Junyang Lin. The Lessons of Developing Process Reward Models in Mathematical Reasoning, 2025
> >
> > [2]Thang Luong, Dawsen Hwang, Hoang H. Nguyen, Golnaz Ghiasi, Yuri Chervonyi, Insuk Seo, Junsu Kim, Garrett Bingham, Jonathan Lee, Swaroop Mishra, Alex Zhai, Clara Huiyi Hu, Henryk Michalewski, Jimin Kim, Jeonghyun Ahn, Junhwi Bae, Xingyou Song, Trieu H. Trinh, Quoc V. Le, Junehyuk Jung. Towards Robust Mathematical Reasoning, 2025

---

> > > ### Comment · Reviewer_4VNo · 2025-11-20
> > > **Official Comment by Reviewer**
> > >
> > > Thank you for the detailed rebuttal and I find the additional clarifications reasonable. While I still view the empirical gains over the GRPO baseline as modest, the method is a simple extension of existing approaches with demonstrably more stable improvements, and I have updated my score accordingly.

---

> > > > ### Author Response · Authors · 2025-11-20
> > > >
> > > > We appreciate your discussion. Thank you for your thoughtful reconsideration and for updating your score. We’re glad the clarifications were helpful, and we remain happy to address any further questions you may have.

---

### Official Review · Reviewer_6nHT · 2025-11-01

**Soundness:** 3
**Presentation:** 3
**Contribution:** 3
**Rating:** 6
**Confidence:** 4

**Summary:**

Authors demonstrate Lean proof assistant's fine grained feedback on proofs can be harnessed as tactic level process rewards that produce improved RL training results. They compare outcome only, process only, and their hybrid method and show improved training dynamics. They demonstrate solid results on miniF2F and ProofNet and contribute detailed ablation studies for several design decisions.

**Strengths:**

- thorough definitions and descriptions of their procedures
- clean idea that leverages the additional signal afforded by powerful proof assistants
- thorough ablation studies confirming the significance of various design decisions

**Weaknesses:**

- analysis limited to full proof generation-- search and iterative repair remain important techniques in automatic theorem proving
  - this point is slightly biased towards the human perspective, but theorem proving is often understood as a search problem

- nit: reward definition hard codes values given to "valid steps" before incorrect steps. This measure of validity is only in the syntax sense, right? The tactic is not necessarily contributing to a correct proof. The empirical analysis with d1/d2 shows that it should be treated distinctly from syntactically incorrect tactics which makes sense, but beyond this ordering, selecting a fixed value doesn't seem grounded.

**Questions:**

- has there been any opportunities to further scale training? the differences between different conditions seem fairly small on the outcome reward chart in figure 2, and RL is infamously unstable (even in the R1 paper poor trends (negative/plateau) are observable for hundreds of steps before another rise). I totally understand if it's a resource limitation, but curious to see if the patterns highlighted here hold at scale.

---

> ### Author Response · Authors · 2025-11-19
>
> We thank the reviewer for raising this important point.
>
> ### **Q1) Comparison between Search-based method and Whole-Proof Generation**
>
> We agree that some LLM-based theorem provers are formulated as search problems [1, 2].
>
> In contrast, a parallel line of work employs a **whole-proof generation** paradigm [3, 4], where the model produces an entire Lean proof in a single forward pass, and Lean verifies it afterward. This approach avoids inference-time search or repair and isolates the behavior of the proof generator itself.
>
> In this paper, we deliberately adopt the whole-proof generation setting used in [3,4] in order to cleanly study the effect of **Lean-derived tactic rewards on training time credit assignment**, without confounding factors such as search depth, branching heuristics, or iterative repair strategies at inference time. As a result, our analysis does not employ search-based decoding, and thus we do not provide search-based qualitative examples.
>
> We view our method as **complementary** to search frameworks rather than a replacement. Integrating Lean-guided tactic rewards with search or iterative repair is an interesting extension, and we will clarify this positioning and scope in the revision.
>
> ### **Q2) On reward definition**
>
> We appreciate the opportunity to clarify this point. In our setting, a “valid” tactic is not merely syntactically well-formed. Lean’s elaborator checks each tactic against the *current* proof state under dependent type theory: it verifies whether the tactic is *locally* logically sound and legally applicable in the given context. A tactic that produces no error in the info tree is therefore locally type-theoretically sound, even if the overall proof ultimately fails. This is considerably stronger than a pure syntax check and represents genuine semantic correctness at the step level.
>
> Regarding the fixed values d_1 and d_2: as noted in our Limitations section, using static scores is indeed a design restriction. However, our ablation studies show that the learning dynamics and final performance are **robust across a wide range of (d₁, d₂) choices**, and that the key effect arises from distinguishing (i) globally correct proofs, (ii) locally valid tactics, and (iii) erroneous tactics—rather than from the specific numeric values.
>
> **Furthermore, we added Appendix J, which provides a mathematical interpretation of our reward scheme. Specifically, we show that (a) under a first error propagation assumption, an ideal value function exhibits a monotone structure, and (b) our discrete tactic rewards can be formulated as a potential-based shaping term consistent with that structure.**
>
> ### **Q3) Scale Experiment**
>
> We fully acknowledge the reviewer’s point. However, in our current setting, we are constrained by available compute and by the model sizes used in formal theorem proving. Most existing Lean-based provers occupy two extremes: relatively small models around 7B parameters (such as the models we used in the paper), and very large models such as DeepSeek-Prover-V2 (671B) or Kimina-Prover (72B). Training or RL-fine-tuning models at the latter scale with Lean-in-the-loop verification is beyond our current computational budget. We view extending this framework to larger provers (e.g., 70B+) as important future work and will make these resource limitations more explicit in the revised version.
>
> [1] Zijian Wu, Suozhi Huang, Zhejian Zhou, Huaiyuan Ying, Jiayu Wang, Dahua Lin, and Kai Chen.
> Internlm2.5-stepprover: Advancing automated theorem proving via expert iteration on large-scale
> lean problems, 2024
>
> [2] Haohan Lin, Zhiqing Sun, Sean Welleck, and Yiming Yang. Lean-star: Learning to interleave
> thinking and proving, 2025
>
> [3] Haiming Wang, Mert Unsal, Xiaohan Lin, Mantas Baksys, Junqi Liu, Marco Dos Santos, Flood
> Sung, Marina Vinyes, Zhenzhe Ying, Zekai Zhu, Jianqiao Lu, Hugues de Saxcé, Bolton Bailey,
> Chendong Song, Chenjun Xiao, Dehao Zhang, Ebony Zhang, Frederick Pu, Han Zhu, Jiawei Liu,
> Jonas Bayer, Julien Michel, Longhui Yu, Léo Dreyfus-Schmidt, Lewis Tunstall, Luigi Pagani,
> Moreira Machado, Pauline Bourigault, Ran Wang, Stanislas Polu, Thibaut Barroyer, Wen-Ding
> Li, Yazhe Niu, Yann Fleureau, Yangyang Hu, Zhouliang Yu, Zihan Wang, Zhilin Yang, Zhengying
> Liu, and Jia Li. Kimina-prover preview: Towards large formal reasoning models with reinforcement learning, 2025
>
> [4] Jingyuan Zhang, Qi Wang, Xingguang Ji, Yahui Liu, Yang Yue, Fuzheng Zhang, Di Zhang, Guorui Zhou, and Kun Gai. Leanabell-prover: Posttraining scaling in formal reasoning, 2025

---

### Official Review · Reviewer_kG7S · 2025-11-01

**Soundness:** 3
**Presentation:** 3
**Contribution:** 3
**Rating:** 8
**Confidence:** 4

**Summary:**

The paper studies reinforcement learning for formal mathematics in Lean. It proposes to view Lean compiler feedback not just as a binary final reward, but as more fine-grained process supervision: a large penalty is assigned to the tactic that produces the first error message. Continued RL experiments on DeepSeekProver-V1.5 and STP models suggests that this method performs better than standard outcome-only RL.

**Strengths:**

- The method is sensible and clearly described, the experimental setup is sound, the experimental results are good.
- The problem of credit assignment is highly relevant: both in code generation and (natural language + formal) math, the state of the art uses outcome-only reinforcement learning, which forgoes the finer, localized signal that compilation error messages or generative reward models provide.

**Weaknesses:**

- The presentation is overly convoluted (e.g. suggestion: remove TacSet etc, write: "viewing a proof $Y$ as a sequence of tactics $(T_1, ..., T_n)$ parsed from the AST") and lacks crucial details (which values for $d_1$ and $d_2$ are chose? which one is larger?).
- The method could be grounded more theoretically: first come up with a credit assignment scheme, then explain that any token-level advantage sequence can be integrated into GRPO. The optimal way of assigning credit is via $r_t + V(s_{t+1}) - V(s_t)$ where $V$ represents the expected value under the policy $\pi$. Rewards $r_t$ are still sparse at the end of the sequence. Under the modeling assumption "no recovery after the first error" that the authors rightly put forward in the context of Lean, $V(s_{t+1})$ is zero after the first error. $V(s_0)$ is estimated from the $n$ rollouts in GRPO. Inbetween, there is incomplete information to fill in for $V$ precisely, but the constant extrapolation suggested in the paper is likely suboptimal. Under a model of independent Bernoulli-distributed error variables at each tactic, the value $V$ should *increase* over the progression of the trajectory until reaching a value of 1 upon success or until the first error. I do not expect the resulting assignment scheme to produce fundamentally different behavior, but it would be a more theoretically sound derivation of a scheme similar to the one used in the paper. See: potential-based reward shaping.

**Questions:**

- values for $d_1$ and $d_2$?

---

> ### Author Response · Authors · 2025-11-19
>
> ### **W1) Formulation Presentation**
>
> We appreciate your feedback on our formulation. We will revise the sentence to: “We view a proof $Y$ as a sequence of tactics $(T_1, T_2, \dots, T_{N(Y)})$ parsed from the Abstract Syntax Tree (AST) and sorted by their starting positions, where $N(Y)$ denotes the number of tactics in $Y$, which aligns with the LLM's autoregressive generation process.”
> Regarding the hyperparameters $d_1$ and $d_2$, while these were detailed in the Appendix, we have now explicitly stated them in the main text for better visibility. We used $d_1 = -0.05$ and $d_2 = -0.1$ for the main experiments, and the ablation study for these parameters is presented in Section 5.3 and Appendix C.
>
> ### **W2) Mathematical Ground**
>
> We also appreciate your insight regarding the theoretical interpretation. During the rebuttal period, we added a new section (Appendix J) to interpret our tactic-level rewards. We demonstrate that (i) an idealized first-error propagation rule induces a monotone value structure, and (ii) our discrete Lean-based scores function as a coarse potential-based shaping term consistent with this structure. We did not model the value function using a Bernoulli distribution, **as we assume that tactics within the same proof are not independent**; however, the value function remains monotonically increasing as you suggested. We welcome further discussion regarding this point.
> **(For a clearer view of the mathematical details, please kindly refer to Appendix J, the final appendix of the paper.)**

---

> > ### Author Response · Authors · 2025-11-19
> >
> > We provide a conceptual interpretation of our tactic-level rewards using a simple value model and potential-based reward shaping. Our goal is not to claim a formal optimality guarantee, but rather to clarify how the structure of our discrete Lean-based rewards is aligned with an idealized concept of proof success under a “First error propagation” assumption with potential function.
> >
> > **1. Define the Value Function**
> >
> > Consider a Lean proof trajectory
> > $s_0 \xrightarrow{T_1} s_1 \xrightarrow{T_2} \cdots \xrightarrow{T_N} s_N,$
> >
> > where $s_t$ denotes the prefix of tactics ($T_1,\dots,T_{t-1}$), and $s_N$ refers to a completed proof.
> >
> > We adopt the modeling assumption already used in the main paper:
> >
> > From the first error propagation, once the first erroneous tactic occurs, the proof can no longer be repaired into a valid Lean proof.
> >
> > Formally, let $j$ be the index of the first tactic for which Lean reports an error. Then all states $s_t$ with $t\ge j$ lie in an absorbing “failed” state.
> > Instead of assuming independent Bernoulli errors, we consider a more general and realistic model with conditional valid probabilities. For a valid prefix $s_{k-1}$, let
> > $q(s_{k-1}) = P(\text{no error at step }k \mid s_{k-1}\text{ is valid}).$
> >
> > Under the first error propagation assumption, we define the value function as the probability of eventually producing a valid proof from a valid prefix $s_t$ is
> > $V(s_t)= P(\text{success} \mid s_t)= \prod_{k>t} q(s_{k-1}).$
> >
> > Along a valid trajectory, we have
> > $V(s_{t+1}) = \frac{V(s_t)}{q(s_t)} \ge V(s_t),$
> >
> > so $V(s_t)$ is monotone increasing until no errors are founded. if the first error occurs at step $j$, the success probability collapses to zero:
> > $V(s_j) = V(s_{j+1}) = \cdots = 0.$
> >
> > Qualitatively, this yields the following structure:
> > - For a successful proof, $V(s_t)$ increases from a small value at $s_0$ to $V(s_N)=1$.
> > - For a failed proof, $V(s_t)$ increases along the correct prefix, and then drops to 0 at the first error and stays at 0 afterwards.
> >
> >
> > Thus, the “ideal” value function encodes (i) Monotone growth along error-free prefixes and (ii) Irreversible collapse after the first error.
> >
> > This structure motivates using stronger positive credit for tactics on an error-free prefix and negative or neutral credit after the first failure.
> >
> >
> > **2. Potential-Based Shaping with $V(s)$**
> >
> > The value defined in previous section suggests a way to define potential-based shaping. In an idealized MDP setting where the environment state is exactly \(s_t\) and the agent has access to the value function $V(s)$, one could define a potential
> >
> > $\Phi^\star(s) = f\bigl(V(s)\bigr),$
> > where $f$ is any monotonically increasing transformation (e.g., $f(v)=v$).
> > A shaped reward can then be written as
> >
> > $r_t^\star \=\ r_{\text{outcome},t} + \gamma \Phi^\star(s_{t+1}) - \Phi^\star(s_t),$
> >
> > where $r_{\text{outcome},t}$ is the sparse end-of-proof reward derived from $g(Y)$. Under the assumptions of Ng et al. (1999), such potential-based shaping preserves the set of optimal policies.
> >
> > Intuitively, using $\Phi^\star(s) = f(V(s))$ means that the potential is highest on globally successful trajectories, increases along error-free prefixes, and collapses after the first error, similar with the structure of $V(s)$ in previous section. The corresponding temporal-difference term
> >
> > $\gamma \Phi^\star(s_{t+1}) - \Phi^\star(s_t)$
> > acts as a local improvement signal: it is positive along valid contexts,  negative when the value collapses at the first error, and zero afterwards.
> >
> > In our setting, however, we neither assume access to the true $V(s)$.  We therefore view this potential-based construction as a normative model that suggests the qualitative shape of a desirable local credit signal, rather than as a source of formal optimality guarantees.

---

> > > ### Author Response · Authors · 2025-11-19
> > >
> > > **3. Lean-based Discrete Approximation as Quantized Local Shaping**
> > >
> > > In practice, we did not estimate $V(s)$ or $\Phi^\star(s)$ explicitly.
> > > Instead, we exploit Lean's symbolic feedback (AST errors, first-error propagation) to construct discrete tactic-level scores.
> > >
> > > For a proof $Y$, with first error index $j$ (if any), recall that we define
> > >
> > > We define $\phi(Y, T_t)$ as follows:
> > >
> > > • $\phi(Y, T_t) = 1$ if $g(Y)=1$;
> > >
> > > • $\phi(Y, T_t) = d_1$ if $g(Y)=0$ and $t<j$;
> > >
> > > • $\phi(Y, T_t) = d_2$ if $g(Y)=0$ and $t\ge j$,
> > >
> > > with $1 > d_1 > d_2$.
> > >
> > > as the process-level reward for tactic $T_t$.
> > >
> > > Conceptually, $\varphi(Y,T_t)$ is a coarse, Lean-driven \emph{quantization} of the ideal local improvement signal suggested by the value model. States on globally successful trajectories receive the highest score ($1$); states on error-free prefixes of failed proofs receive an intermediate score ($d_1$); and states at or after the first error receive the lowest score ($d_2$). This partitions trajectories into three regions whose ordering (success $>$ pre-error $>$ post-error) is aligned with the ordering of $V(s)$ implied by previous section.
> > >
> > > Assuming $\gamma=1$ and a finite horizon, any such per-step process reward
> > > sequence can be written as a difference of a state potential. For a fixed
> > > trajectory $Y$ of length $N$, define $\Phi$ backwards by
> > >
> > > $
> > > \Phi(s_N) = 0,\qquad
> > > \Phi(s_{t+1}) - \Phi(s_t) = r_{\mathrm{process},t},\quad t=N-1,\dots,0.$
> > >
> > > By construction,
> > >
> > > so the total shaped reward becomes
> > >
> > > r'*t = r_outcome,t + r_process,t = r_outcome,t + Φ(s*{t+1}) - Φ(s_t)
> > >
> > > This potential $\Phi$ is not intended as an estimate of the true value function $V(s)$; rather, it is an implicit potential induced by our discrete Lean-based scoring rule. The key point is that its level sets respect the same qualitative ordering (success $>$ pre-error $>$ post-error) as the ideal value model, providing a theoretically motivated yet practical shaping signal.
> > >
> > > **4. Discussion and Limitations**
> > >
> > > Our analysis suggests the following:
> > >
> > > 1)Under a first-error propagation assumption, an ideal value function $V(s)$ for Lean proofs increases along error-free prefixes and collapses to zero after the first error.
> > >
> > > 2)Our discrete tactic-level scores $\varphi(Y,T_t)\in{1,d_1,d_2}$can be viewed as a quantized local improvement signal, capturing this qualitative structure without estimating $V(s)$ explicitly.
> > >
> > > 3)For any given trajectory, the resulting process rewards $r_{\mathrm{process},t}$ can be written as a potential-based shaping term $r_{\mathrm{process},t} = \Phi(s_{t+1}) - \Phi(s_t)$ for a suitable potential $\Phi$.
> > >
> > > Consequently, our use of potential-based shaping should be understood as a theoretical framework that explains the structure of our rewards and motivates our design choices, rather than as a strict proof that our procedure preserves the optimal policy for the original sparse outcome reward. Empirically, we observe that this verifier-informed, discretized shaping leads to more stable training and consistent improvements over outcome-only GRPO on MiniF2F and ProofNet. We emphasize that we do not claim any formal optimality guarantee for this shaped reward in our large-scale LLM–Lean setting; the potential-based perspective is used purely as a conceptual framework for designing and interpreting our tactic-level rewards.

---

### Meta-Review · Area_Chair_mVhn · 2026-01-07

**Summary:**

This paper proposes using the Lean proof assistant as a symbolic process oracle during reinforcement learning training for theorem proving. The key idea is to parse generated proofs into tactic sequences and leverage Lean's elaboration feedback to assign tactic-level rewards in addition to outcome-level rewards.

The paper was well received, with almost all reviewers finally agreeing towards acceptance of the paper. Therefore I recommend acceptance. That said, the gain of the proposed approach is very tiny over normal outcome-reward RL, and it is not clear if the method is very useful in the long term or not.

**Reviewer Concerns:**

Several reviewers initially questioned whether the contribution amounted to a straightforward application of proof-assistant feedback or whether it meaningfully differed from existing approaches such as PRMs. In response, the authors clarified that the core contribution lies in formulating an online RL objective that bridges symbolic, tactic-level verifier signals with token-level credit assignment. This involved formalizing a tactic-level MDP, designing reward shaping and first-error propagation mechanisms, and reducing symbolic feedback to training-compatible scalar rewards. These clarifications were received positively, with at least one reviewer explicitly indicating an intent to raise their score.

Additional concerns focused on the role of proof-assistant feedback relative to whole-proof generation, the nature of the verifier signals being exploited, and scalability. The authors explained their deliberate choice of whole-proof generation to isolate training-time credit-assignment effects, emphasized that proof assistants provide local logical validity grounded in dependent type theory rather than mere syntactic checks, and acknowledged current scalability limitations due to the lack of accessible large-scale formal-proof models. Reviewers responded positively to these explanations, and one reviewer updated their score upward during the discussion.

Finally, in response to feedback on theoretical grounding, the authors added a formal value-function and potential-based interpretation of their reward mechanism.

Overall, the reviewers were happy with the clarifications.

**Reviewer Scores:**

I think it would have been 6, 6, 6, 8.

---

### Decision · Program_Chairs · 2026-01-26

Accept (Poster)